# Nonparametric LLM Evaluation from Preference Data

**Dennis Frauen** [1 2]  **Athiya Deviyani** [3]  **Mihaela van der Schaar** [4]  **Stefan Feuerriegel** [1 2]

## Abstract

Evaluating the performance of large language models (LLMs) from human preference data is crucial for obtaining LLM leaderboards. However, many existing approaches either rely on restrictive parametric assumptions or lack valid uncertainty quantification when flexible machine learning methods are used. In this paper, we propose a nonparametric statistical framework called DMLRANK *for comparing and ranking LLMs from preference data* using debiased machine learning (DML). For this, we introduce *generalized average ranking scores (GARS)*, which generalize commonly used ranking models, including the Bradley-Terry model or PageRank/ Rank centrality, with complex human responses such as ties. DMLRANK comes with the following advantages: (i) It produces statistically efficient estimates of GARS ranking scores. (ii) It naturally allows the incorporation of black-box machine learning methods for estimation. (iii) It can be combined with pre-trained LLM evaluators (e.g., using LLM-as-a-judge). (iv) It suggests optimal policies for collecting preference data under budget constraints. We demonstrate these advantages both theoretically and empirically using both synthetic and real-world preference datasets. In summary, our framework provides practitioners with powerful, state-of-the-art methods for comparing or ranking LLMs for leaderboards.

## 1. Introduction

Large language models (LLM) are commonly evaluated using *preference data*, where models are compared by directly judging which of two generated responses is better for a given prompt. This paradigm is widely used in leaderboards such as *LM Arena* (Chiang et al., 2024) and related

benchmarks for model comparison (Ouyang et al., 2022; Rafailov et al., 2023; Dubois et al., 2023). In practice, collecting preferences is often simpler and more reliable than assigning absolute quality scores to individual LLM outputs (Christiano et al., 2017).

In a typical setup, two LLMs (say, A and B) are prompted with the same task, their generated responses are presented to an evaluator, and the evaluator indicates whether the response of A or B is preferred (see Fig. 1). The evaluator may be a human (e.g., a crowdworker or a domain expert) or an automated system (such as an LLM-as-a-judge). The outcome of this process is typically a leaderboard or ranking of models, where practitioners are often interested not only in the relative ordering of models but also in confidence intervals for the underlying model scores (e.g., as in the case of *LM Arena*) (Chiang et al., 2024).

Despite its practical success, statistical ranking from preference data poses substantial challenges. The main difficulty is that preference data contains only *relative* information: we never directly observe an absolute quality score for a model's output, but only that one response was judged better than another. As a result, this typically requires practitioners to define a notion of a *ranking score* that can be inferred from pairwise comparisons (Bradley & Terry, 1952). These ranking scores are often complex functionals of the data distribution, which makes statistical inference challenging. Hence, estimation procedures must be carefully tailored to the ranking target of interest to guarantee efficient estimation and valid uncertainty quantification (Frick et al., 2025; Kallus, 2025).

Many existing LLM leaderboards and other preference-based benchmarks build on the parametric Bradley–Terry (BT) model, which posits that each LLM's underlying ranking score determines the preference probabilities additively on the logit scale (Chiang et al., 2024; Frick et al., 2025). However, the parametric BT model comes with important **limitations**: *(i)* It can be *biased under misspecification*, such as when the BT link is incorrect for the data-generating process or if the chosen parametric model is too restrictive (Xu et al., 2025a); *(ii)* The BT is parametric and generally *not* compatible with flexible machine learning (ML) estimators that model context-dependent preferences, as this typically breaks standard inferential guarantees and can lead to invalid

[1]LMU Munich [2]Munich Center for Machine Learning [3]Carnegie Mellon University [4]University of Cambridge. Correspondence to: Dennis Frauen <frauen@lmu.de>.

*Proceedings of the 43rd International Conference on Machine Learning*, Seoul, South Korea. PMLR 306, 2026. Copyright 2026 by the author(s).

confidence intervals (van der Vaart, 1998; Chernozhukov et al., 2018). *(iii)* Many modern evaluation pipelines combine scarce but high-quality human labels with proxy labels from auto-raters or LLM-as-a-judge systems (Zheng et al., 2023). However, for parametric models, incorporating such proxy signals while obtaining valid statistical inference is *not* straightforward.

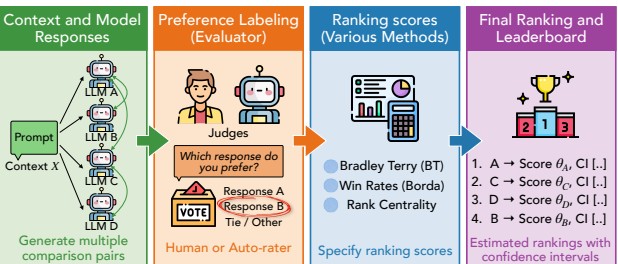

*Figure 1.* **Overview of preference-based LLM evaluation.**

In this paper, we propose a *nonparametric* statistical framework called DMLRANK to compare and rank LLMs from preference data while addressing the above limitations. Our key idea is to define the ranking target directly as a functional of the contextual preference probabilities, which we call *generalized average ranking scores (GARS)*. GARS naturally accommodate different notions of ranking scores such as average win rates (Borda scores), stationary-distribution scores (Rank Centrality / PageRank-style) (Negahban et al., 2017), or BT-type scores as a special case.

Building on debiased machine learning (DML), we then derive statistically efficient estimators of GARS ranking scores by deriving the corresponding efficient influence function (van der Vaart, 1998). We further construct confidence intervals that remain valid when preference probabilities are estimated with black-box machine learning methods. We also show how DMLRANK can incorporate external judges such as LLM-as-judge, or other pre-trained LLM evaluators. Finally, we also propose an extension for *optimal preference data acquisition* by deriving collection policies that minimize the width of the resulting confidence intervals under budget constraints, thereby enabling cost-efficient preference data collection.

Our main **contributions**[1] are: (i) We propose a generalized nonparametric framework for statistical ranking from preference data. Specifically, we introduce *generalized average ranking scores (GARS)*, a nonparametric estimand that captures many existing ranking models. (ii) We develop a statistically efficient estimator for GARS, which directly yields simultaneously valid confidence intervals for ranking scores. (iii) We propose a cost-optimal policy for collecting preference labels under budget constraints.

---

[1]Code is available at https://github.com/DennisFrauen/NonparametricLLMEval.

**Conflict of Interest Disclosure.** The authors declare no financial conflicts of interest related to this work. In particular, no author has a financial interest that could reasonably be perceived as influencing the research, results, or conclusions of this paper.

## 2. Related work

We review the main related literature streams on debiased machine learning and LLM evaluation. We refer to Appendix A for details on optimal experimental design.

**Debiased machine learning.** Debiased/double machine learning (DML) has a long tradition in semiparametric statistics (Bickel et al., 1998). DML leverages the so-called efficient influence function (EIF) (Hampel, 1974) to obtain semiparametrically efficient estimators (van der Vaart, 1998) or Neyman-orthogonal risk functionals (Foster & Syrgkanis, 2023). DML is generally useful when the estimation target is defined via unknown nuisance components (Chernozhukov et al., 2018). Examples of such problems include missing data analysis (Robins et al., 1994), semi-supervised learning (prediction-powered inference) (Angelopoulos et al., 2023; 2024; Xu et al., 2025b), and causal inference (Robins et al., 2000; van der Laan & Rubin, 2006; Nie & Wager, 2021; Kennedy, 2023). Our work brings these ideas to preference-based evaluation using various ranking scores.

**Semi-supervised model evaluation.** A growing literature combines small amounts of high-quality labels with large unlabeled data or cheap proxy labels. For instance, Shanmugam et al. (2025) studied semi-supervised evaluation for standard labels, but *not* preference data, and *without* DML. Another literature stream builds upon prediction-powered inference (PPI) to develop debiased methods that leverage black-box predictors to reduce labeling cost (Fisch et al., 2024; Guerdan et al., 2026). Finally, Chatzi et al. (2024) uses PPI for preference-based ranking with uncertainty quantification *but* focuses on specific ranking definitions. In contrast, we provide a unified EIF-based framework covering multiple ranking estimands (including BT-type, win-rate/Borda, and stationary-distribution scores) and also address optimal data collection.

**LLM evaluation from preference data.** LLM leaderboards, such as *LM Arena*, have popularized *pairwise* comparison data as a practical evaluation signal. The same paradigm has also been used in specialized domains, including search-augmented models (Miroyan et al., 2026), text-to-music (Kim et al., 2025), and coding assistants (Chi et al., 2025). However, current methods for estimating ranking scores rely on the *parametric* BT model. As a consequence, these methods are either *sensitive to misspecification* or do *not* yield valid asymptotic confidence intervals if flexible ML models are used. In parallel, simulation- and

benchmark-driven pipelines such as AlpacaFarm/AlpacaEval (Dubois et al., 2023; 2024) commonly employ nonparametric win-rate or average-pairwise-win estimands, but are typically treated without a debiased efficiency perspective. Hence, the underlying ranking methods are *neither* statistically efficient *nor* yield principled, efficient uncertainty quantification when black-box ML models are used.

**Semiparametric ranking.** There is extensive work on semiparametric estimation for discrete choice and comparison models in econometrics (Horowitz, 1992; Klein & Spady, 1993). These works typically focus on identifying and estimating low-dimensional preference parameters rather than providing a unified inference framework for *arbitrary* ranking functionals of high-dimensional, context-dependent preference probabilities, and they generally do *not* address missing labels or the use of auxiliary judges. More closely related, several recent papers study semiparametric inference and optimization for preference learning and LLM alignment under BT-style models (Sawarni et al., 2025; Kallus, 2025). Two recent works derive inference procedures for semiparametric/covariate-adjusted BT models (Li & Li, 2025; Spokoiny, 2025). In contrast to this paper, these approaches remain tied to BT-type score models and do *not* offer a unified treatment of alternative ranking functionals (e.g., Borda/win-rate or Rank Centrality), and do *not* address optimal preference data acquisition.

## 3. Problem setting

**Setting.** Our aim is to rank $K$ items in $\mathcal{I} = \{1, \dots, K\}$ using preference data with $C$ different categories $\mathcal{C} = \{1, \dots, C\}$. Given an item pair $(j, k)$ with $j, k \in \mathcal{I}$, a common example is three categories $\mathcal{C} = \{1, 2, 3\}$ with categories labeled as *"1: j preferred over k," "2: k preferred over j,"* or *"3: tie (no preference)."* The data consists of a context $X \in \mathcal{X}$, binary selection indicators $S_{jk} \in \{0, 1\}$ with zero diagonals $S_{jj} = 0$ and $S_{jk} = 1$ if the item pair $(j, k)$ is labeled, and one-hot encoded categorical preference labels $Y_{jk} \in \{e_1, \dots, e_C\}$, where $Y_{jk} = e_c \in \{0, 1\}^C$ denotes assignment of category $c$ to the pair $(j, k)$. Finally, we note that we allow for selecting multiple pairs of items per context $X$, but we also discuss the case of selecting one or fewer pairs per context in Appendix G.

**Running example: LLM evaluation.** Here, the set of items $\mathcal{I}$ is the different LLMs that we want to compare using human feedback. The context $X$ is a prompt that is given to pairs of LLMs $(j, k)$ selected for comparison. $S_{jk}$ indicates whether human preference was acquired and $Y_{jk}$ is the human preference category. As an extension, we (optionally) may have access to a black-box model $f(x, j, k)$ that predicts the preference $Y_{jk}$ given a context $X = x$ and items $(j, k)$. We make no assumptions on the model $f$ (i.e., on its prediction performance). The black-box model $f$ may

be an auto-rater or an LLM-as-a-judge.

**Observed data.** We write the observed data as $O = (X, S, \widetilde{Y})$ with $S \in \{0, 1\}^{K \times K}$ and $Y \in \{0, 1\}^{K \times K \times C}$ denoting stacked tensors of selection indicators and preference labels, and $\widetilde{Y} = S \circ Y$ such that $Y_{jkc}$ is only observed for all $c \in \mathcal{C}$ whenever $S_{jk} = 1$. We assume that the population $(X, S, \widetilde{Y}) \sim \mathbb{P}$ follows some ground-truth distribution $\mathbb{P}$ and that we have access to a *preference dataset* $\mathcal{D} = \{(x_i, s_i, \widetilde{y}_i)_{i=1}^n\}$ sampled i.i.d. from $\mathbb{P}$.

**Nuisance functions.** We denote

$$\mu_{jkc}(x) = \mathbb{P}(Y_{jkc} = 1 \mid X = x) \tag{1}$$

as probability of preference category $c$ for pair $(j, k)$ given context $x$. Furthermore, we denote

$$\pi_{jk}(x) = \mathbb{P}(S_{jk} = 1 \mid X = x) \tag{2}$$

as the selection probability of pair $(j, k)$ given context $x$. We define $\mu(x) \in [0, 1]^{K \times K \times C}$ and $\pi(x) \in [0, 1]^{K \times K}$ as the corresponding stacked tensors with convention $\mu_{jjc}(x) = \pi_{jj}(x) = 0$ for all $j, c$. We denote the collection $\eta = (\mu, \pi)$ as *nuisance functions*. We make the standard assumptions of positivity and missing at random (Rubin, 1974), under which we can write

$$\mu_{jkc}(x) = \mathbb{P}(Y_{jkc} = 1 \mid X = x, S_{jk} = 1) \tag{3}$$

(see Appendix B.1).

**BT model.** The *Bradley-Terry (BT) model* (Bradley & Terry, 1952) is commonly used ranking model in the simplified case of binary preference categories $\mathcal{C} = \{1, 2\}$, where $Y_{jk1} = 1$ denotes preference of $j$ over $k$ (and vice versa for $Y_{jk2} = 1$). BT is defined via

$$\mu_{jk1}(x) = \sigma\big(r_j(x) - r_k(x) + b(x)\big) \tag{4}$$

with the convention $\sum_{m=1}^{K} r_m(x) = 0$ and where $\sigma(t) = \frac{1}{1+e^{-t}}$ denotes the sigmoid function. The vector $r(x) = (r_1(x), \dots, r_K(x))$ denotes the *ranking score functions* of the items, and $b(x)$ models the position bias due to showing items in a certain order. The target of interest is the average *ranking score* vector $\theta = \mathbb{E}[r(X)] \in \mathbb{R}^K$, which allows ranking of items according to the components of $\theta$. Popular leaderboards such as *LM Arena* (Chiang et al., 2024; Frick et al., 2025) leverage *parametric* BT models to perform statistical inference on the target $\theta$. That is, the score function $r_\rho(x)$ is parametrized with some trainable parameters $\rho$ (e.g., a neural network). An estimator $\hat{\rho}$ can be obtained via maximum likelihood estimation, and the target parameter can be estimated via

$$\hat{\theta} = \frac{1}{n} \sum_{i=1}^{n} r_{\hat{\rho}}(x_i). \tag{5}$$

**Research questions.** We address *three main questions*:

> ❶ **Nonparametric ranking scores** (→Sec. **4**). How can we define a vector of ranking scores $\theta \in \mathbb{R}^K$ *nonparametrically* only in terms of the nuisance functions $\eta$? How does $\theta$ generalize established notions of ranking scores such as BT?
>
> ❷ **Efficient statistical inference** (→Sec. **5**). How can we leverage preference dataset $\mathcal{D}$ and an (optional) black-box judge $f$ in the most efficient way to perform statistical inference on the ranking scores $\theta$ (estimator and confidence intervals with black-box machine learning)?
>
> ❸ **Optimal data acquisition** (→Sec. **6**). How can we design an optimal labeling rule $\pi$ to obtain preference data for optimal inference on $\theta$ subject to budget constraints (e.g., costly human annotations)?

## 4. Generalized Average Ranking Scores

### 4.1. Motivation

**Why parametric approaches have drawbacks.** Parametric approaches such as the parametric BT model have multiple *drawbacks*: (i) the parametric estimator will generally be biased if the parametric model $r_\rho(x)$ is misspecified (i.e., the model class chosen is too small); (ii) it is generally difficult to obtain valid (asymptotic) confidence intervals for $\hat{\theta}$ if flexible models such as neural networks are used for $r_\rho(x)$ (Chernozhukov et al., 2018); and (iii) it is unclear how pre-trained evaluation models (e.g., auto-rater of LLM-as-a-judge) can be incorporated into this procedure.

**Idea behind our nonparametric approach.** Motivated by the drawbacks of the parametric BT model, we propose a *nonparametric* modeling approach. The key idea is to express the target parameter $\theta$ as a function of the preference probabilities $\mu$, that is, $\theta = \mathbb{E}[F(\mu(X))]$ for some known function $F$. Then, we could use any black-box machine learning method or pre-trained auto-rater to obtain estimated preferences $\hat{\mu}(x)$ and use these to obtain an estimator $\hat{\theta}$.

Indeed, it is possible to construct such a function $F$ as follows: Let $B \in \mathbb{R}^{\binom{K}{2} \times K}$ be the incidence matrix (row $(j, k)$ equals $e_j - e_k$) and let $L_0 = B^\top B$ be the graph Laplacian. Let $\ell(\mu(x)) = (\text{logit}(\mu_{jk1}(x))/2 + \text{logit}(\mu_{kj2}(x))/2)_{j<k} \in \mathbb{R}^{\binom{K}{2}}$ and let

$$H = \begin{bmatrix} I_{K-1} \\ -\mathbf{1}_{K-1}^\top \end{bmatrix}, \qquad (6)$$

so that $\{r \in \mathbb{R}^K : \mathbf{1}^\top r = 0\} = \{H\alpha : \alpha \in \mathbb{R}^{K-1}\}$. We can then define the function $F$ via

$$F(\mu(x)) = H (H^\top L_0 H)^{-1} H^\top B^\top \ell(\mu(x)), \quad (7)$$

which coincides with the $L^2$-projection of the edge log-odds vector $\ell(\mu(x))$ onto the subspace $\{B\phi : \mathbf{1}^\top \phi = 0\}$ and recovers the BT-scores $\mathbb{E}[r(X)]$ if the BT model from Eq. (4) holds (see Appendix D).

**Ranking beyond BT.** The BT model is a convenient and widely used model for pairwise comparisons, but it is *not* universally appropriate. In practice, it can be misspecified and yield biased rankings if, e.g., preferences exhibit cycles (Xu et al., 2025a). This has motivated a broad literature on alternative definitions of ranking scores. As emphasized in social choice theory, no single ranking method is optimal across all desiderata; different models trade off interpretability, robustness, and other desired properties (Zhang & Hardt, 2024). Motivated by this insight, we now develop a generalized framework that accommodates a range of ranking score definitions within one common inferential framework.

### 4.2. Formulation of GARS

In this paper, we propose *generalized average ranking score (GARS)*, which we define as follows: Given a known function $F: [0, 1]^{K \times K \times C} \to \mathbb{R}^d$ for some dimension $d > 0$, GARS is defined as the parameter vector

$$\theta = \mathbb{E}[F(\mu(X))] \in \mathbb{R}^d. \qquad (8)$$

**Examples.** In the following, we consider examples for GARS with binary preference categories $\mathcal{C} = \{1, 2\}$. For additional examples, including ties, we refer to Appendix C.

• **Bradley–Terry projections.** as described in Sec. 4.1.

• **Borda scores.** Borda scores are defined as the average probability of an item $j$ to be preferred over the other items:

$$F_j(\mu(x)) = \frac{1}{2(K-1)} \sum_{k \neq j} (\mu_{jk1}(x) + \mu_{kj2}(x)). \quad (9)$$

• **Rank centrality (RC)** (Negahban et al., 2017). Here, we first construct a stochastic matrix from $\mu$; e.g., for $j \neq i$, $T_{ij}(\mu(x)) = \frac{\mu_{ji1}(x) + \mu_{ij2}(x)}{\sum_{\ell \neq i} \mu_{\ell i1}(x) + \mu_{i\ell2}(x)}$, with zero diagonal $T_{ii}(x) = 0$. We define the function

$$F(\mu(x)) = \left(I - T(\mu(x))^\top + \mathbf{1}\mathbf{1}^\top\right)^{-1} \mathbf{1}, \qquad (10)$$

where we assume matrix invertibility. The resulting $F(\mu(x))$ is the stationary distribution of the associated Markov chain with transition matrix $T$. Intuitively, human raters randomly switch between LLMs according to their symmetrized preference probabilities. The stationary distribution $F(\mu(x))$ can then be interpreted as the long-term time spent on each LLM (larger implying better).

# 5. Efficient statistical inference

## 5.1. Overview

**Why naïve, plug-in estimators have drawbacks.** To obtain rankings, our goal is now to develop an estimator $\hat{\theta}$ of the GARS $\theta = \mathbb{E}[F(\mu(X))]$ based on the preference dataset $\mathcal{D}$ and an optional pre-trained judge $f$. As outlined in the previous section, our non-parametric framework allows us to estimate $\theta$ only by obtaining an estimator $\hat{\mu}$ of the preference probabilities $\mu$. The natural plug-in estimator is the following: in a first step, one can leverage any black-box machine learning classifier to fit $\hat{\mu}$ on $\mathcal{D}$ or simply use the judge predictions $\hat{\mu}_{jk}(x) = f(x, j, k)$ (we will discuss more sophisticated combinations of these two approaches later). Then, in a second step, one can estimate the GARS via the *plug-in estimator*

$$\hat{\theta} = \frac{1}{n} \sum_{i=1}^{n} F(\hat{\mu}(x_i)). \tag{11}$$

Unfortunately, the plug-in estimator from Eq. (11) has two fundamental drawbacks: (i) It is well known that plug-in estimators suffer from so-called *plug-in bias* (Kennedy, 2024). If errors in estimating $\hat{\mu}$ (e.g., due to limited preference data or due to bias in the judge $f$), these errors will propagate through $F$ to the final estimator $\hat{\theta}$, leading to biased and suboptimal statistical inference. (ii) Due to the plug-in bias, it is generally difficult to obtain valid (asymptotic) confidence intervals for $\hat{\theta}$. For example, if the judge $f$ is fundamentally biased, the plug-in estimator does not correct for this bias and is thus unable to obtain intervals with valid coverage.

**Debiased estimators.** To remedy the drawbacks of the plug-in estimator above, we propose an estimator based on *semiparametric efficiency theory* (van der Vaart, 1998). Specifically, we build on the idea of *debiased estimators*, which correct for the plug-in bias by adding a debiasing term. In our setting, we propose the use of a debiased estimator

$$\hat{\theta} = \frac{1}{n} \sum_{i=1}^{n} F(\hat{\mu}(x_i)) + \alpha \Big(y_i - \hat{\mu}(x_i)\Big), \tag{12}$$

where $\alpha$ is an appropriate scaling matrix that we have to derive. Intuitively, if $\hat{\mu}$ is estimated with error, the second term uses the labeled data to reduce the error from the plug-in term. Debiased estimators will often guarantee statistical efficiency and valid confidence intervals *even if we use black-box ML to estimate $\hat{\mu}$* (Chernozhukov et al., 2018).

## 5.2. Main result

**Deriving the debiased estimator.** We now derive the correct debiased estimator for $\theta$. This is connected to obtaining the so-called *efficient influence function (EIF)* of $\theta$ (Bickel et al., 1998). Importantly, we also show in the following

theorem that our estimator allows for (asymptotic) statistical efficiency (i.e., the lowest variance any unbiased estimator can achieve) and asymptotic normality (leading to valid confidence intervals) under weak assumptions.

**Theorem 5.1** (Efficient statistical inference for GARS). *Let $F : [0,1]^{K \times K \times C} \to \mathbb{R}^d$ be differentiable with Jacobian columns $J_{jk}(\mu) = \nabla_{\mu_{jk}} F(\mu) \in \mathbb{R}^{d \times C}$. The efficient influence function (EIF) for $\theta$ is*

$$\phi(O, \eta, \theta) = F\big(\mu(X)\big) - \theta \tag{13}$$
$$+ \sum_{j \neq k} \frac{S_{jk}}{\pi_{jk}(X)} J_{jk}\big(\mu(X)\big) \Big(Y_{jk} - \mu_{jk}(X)\Big) \in \mathbb{R}^d,$$

*The (one-step) debiased estimator $\hat{\theta}_{\mathrm{EIF}} \in \mathbb{R}^d$ is given via*

$$\hat{\theta}_{\mathrm{EIF}} = \frac{1}{n} \sum_{i=1}^{n} \Bigg[ F\big(\hat{\mu}(x_i)\big) \tag{14}$$
$$+ \sum_{j \neq k} \frac{s_{i,jk}}{\hat{\pi}_{jk}(x_i)} J_{jk}\big(\hat{\mu}(x_i)\big) \Big(y_{i,jk} - \hat{\mu}_{jk}(x_i)\Big) \Bigg],$$

*Under standard assumptions (see Appendix B for details), $\hat{\theta}_{\mathrm{EIF}}$ is asymptotically efficient and normally distributed via*

$$\sqrt{n}(\hat{\theta}_{\mathrm{EIF}} - \theta) \to \mathcal{N}_d(0_{\mathbb{R}^d}, \Sigma), \tag{15}$$

*where $\Sigma = \mathbb{E}\left[\phi(O, \eta, \theta)\phi^\top(O, \eta, \theta)\right] \in \mathbb{R}^{d \times d}$.*

*Proof.* See Appendix B. $\square$

**Interpretation.** The debiased estimator from Eq. (14) weights the individual debiasing terms with the selection probability $\pi_{jk}$ to increase bias correction for rarely observed item pairs. This is similar to the AIPTW estimator in causal inference (Robins, 1994). Different however is the re-weighting using the Jacobian $J_{jk}$, which adjusts strength the bias correction according to how strongly the preference probability affects the target $\theta$ via $F$. In the extreme case where $\theta$ does not depend on $\mu_{jk}$, the Jacobian will be zero, and no bias correction is applied. The Jacobians $J_{jk}$ often admit closed-form solutions and thus allow for scalable computation (see Appendix E).

## 5.3. Implementation

We now present a practical *two-stage procedure* (see Fig. 2) that turns Theorem 5.1 into an estimator with valid confidence intervals.

Stage ①: Nuisance estimation with cross-fitting. We need estimates $\hat{\mu}$ of the preference probabilities $\mu$ and, if unknown, estimates $\hat{\pi}$ of the selection propensities $\pi$. In some scenarios (e.g., when designing preference data as in Sec. 6), $\pi$ may be known in which case we can set $\hat{\pi} = \pi$. As it is standard in semiparametric inference, we use *cross-fitting*

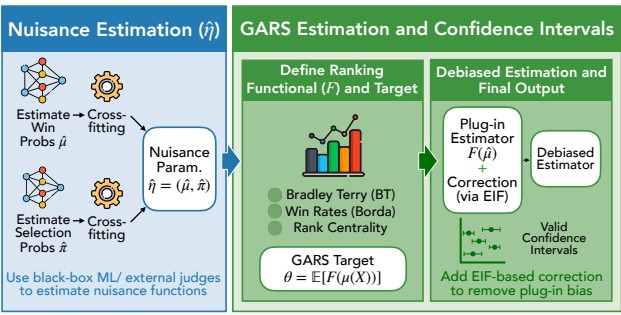

*Figure 2.* **Overview of statistical inference for GARS.**

(Chernozhukov et al., 2018) to obtain $\hat{\mu}$ and $\hat{\pi}$ while guaranteeing valid statistical inference. We randomly split the data $\mathcal{D}$ into $V \geq 2$ folds $\{\mathcal{D}_v\}_{v=1}^V$. For each fold $v$, we use the out-of-fold data $\{\mathcal{D}'_v\}_{v' \neq v}$ to fit models models $\widehat{\mu}^{(-v)}$ and (if needed) $\widehat{\pi}^{(-v)}$. Then, for each $i \in D_v$, we obtain predictions $\widehat{\mu}(x_i) = \widehat{\mu}^{(-v)}(x_i)$ and $\widehat{\pi}(x_i) = \widehat{\pi}^{(-v)}(x_i)$ on the held-out fold $v$.

**Training the nuisance models.** • *Estimating $\mu$.* Recall $\mu_{jkc}(x) = \mathbb{P}(Y_{jkc} = 1 \mid X = x, S_{jk} = 1)$ with categorical labels $Y_{jk} \in \{0, 1\}^C$ that are only observed when $S_{jk} = 1$. Hence, training $\mu$ can be cast as a $C$-class classification problem trained only on labeled pairs: $\{(x_i, j, k, y_{i,jk}) : i \notin \mathcal{I}_v, \ s_{i,jk} = 1\}$. • *Estimating $\pi$.* If $\pi_{jk}(x) = \mathbb{P}(S_{jk} = 1 \mid X = x)$ is unknown, we estimate it via binary classification on the fully observed pairs $\{(x_i, j, k, s_{i,jk}) : i \notin \mathcal{I}_v\}$. For additional details regarding nuisance model training, we refer to Appendix K.

Stage ②: Estimation and confidence intervals. Using cross-fitted $\widehat{\mu}$, $\widehat{\pi}$, and preference data $\mathcal{D}$, we can now compute the debiased GARS estimator $\hat{\theta}_{\text{EIF}}$ from Eq. (14). We can also estimate the asymptotic covariance matrix

$$\widehat{\Sigma} = \frac{1}{n} \sum_{i=1}^n \phi\left(o_i, \hat{\eta}, \hat{\theta}_{\text{EIF}}\right) \phi^\top\left(o_i, \hat{\eta}, \hat{\theta}_{\text{EIF}}\right). \quad (16)$$

Theorem 5.1 then implies that the set

$$\mathcal{E} = \left\{\vartheta \in \mathbb{R}^d : n\,(\widehat{\theta} - \vartheta)^\top \widehat{\Sigma}^{-1} (\widehat{\theta} - \vartheta) \leq \chi^2_{d,\,1-\alpha}\right\} \quad (17)$$

is an asymptotic $1 - \alpha$ confidence ellipsoid for the unknown GARS $\theta$, where $\chi^2_{d,1-\alpha}$ denotes the $d$-dimensional Chi-Squared quantile. It is also possible to use $\widehat{\Sigma}$ for constructing simultaneously valid confidence intervals for the components (ranking scores) of $\theta$. We refer to Appendix I for additional details.

### 5.4. Incorporating external judges

If a black-box judge prediction $f(x_i, j, k)$ for $\mu_{jk}(x_i)$ is available, we use *judge-as-features*: we add $f(x_i, j, k)$ to the model's input features, i.e., setting $\widehat{\mu}_{jk}^{(-v)}(\widetilde{x}_i)$ with

$\widetilde{x}_i = (x_i, f(x_i, j, k))$. If the judge's prediction is of high quality, the model $\widehat{\mu}_{jk}^{(-v)}$ should learn to leverage it for prediction, thus reducing finite sample error. If, however, the judge's prediction is of low quality, the model $\widehat{\mu}_{jk}^{(-v)}$ should learn to ignore it as an input, thus retaining consistent estimation and valid confidence intervals. Cross-fitting ensures any overfitting of the feature-augmenter is confined to the training folds.

## 6. Optimal preference data acquisition

**Labeling under a budget.** So far, we assumed that existing preference data were generated using some fixed *labeling policy* $\pi : \mathcal{X} \to [0, 1]^{K \times K}$, where the labeling probabilities $\pi_{jk}(x) = \mathbb{P}(S_{jk} = 1 \mid X = x)$, denote the probability of labeling a pair of items $(j, k)$ given context $x$. Now, we consider the problem of *designing* such a labeling policy $\pi$ ourselves, which we then can use to acquire new preferences. In practice, however, acquiring human preference labels for LLM pairs may be costly if we, e.g., query domain experts. Hence, we assume that labeling the item pair $(j, k)$ is associated with a cost $c_{jk} > 0$. We then want our labeling policy $\pi$ to fulfill a *budget constraint*, which captures how many preferences we can afford to collect across different contexts and LLM pairs. That is, we define the set of feasible labeling policies as

$$\Pi_{\alpha,\beta} = \left\{\pi : \mathcal{X} \to [\alpha, 1]^{K \times K} : \mathbb{E}\left[\sum_{j \neq k} c_{jk} \pi_{jk}(X)\right] \leq \beta\right\}$$

for some budget $\beta \geq 0$ and where $\alpha > 0$ is a user-selected threshold to ensure positivity.

**Optimal labeling policies.** Our goal is now to derive an optimal labeling policy among the feasible set. For this, we first need to formalize the notion of an optimal labeling policy. Our approach is as follows: an optimal $\pi$ should ensure that the corresponding debiased GARS estimator from Eq. (14) achieves minimal variance when using the newly collected data (and thus the smallest confidence interval width).

For a dataset generated by a fixed $\pi$, this variance is given by the semiparametric efficiency bound $\Sigma = \Sigma(\pi)$ from Theorem 5.1, which can be written as a function of $\pi$ and also characterizes the best variance *any* regular estimator can achieve. Hence, an optimal $\pi^*$ should minimize some meaningful function of the matrix $\Sigma(\pi)$ over the feasible set $\Pi_{\alpha,\beta}$.

**Definition 6.1.** *A labeling policy $\pi^*$ is A-optimal if*

$$\pi^* \in \arg\min_{\pi \in \Pi_{\alpha,\beta}} \operatorname{tr}\left(\Sigma(\pi)\right), \quad (18)$$

*where* $\operatorname{tr}$ *is the trace operator. That is, an A-optimal labeling policy minimizes the sum of variances of the individual ranking score estimators.*

An A-optimal $\pi^*$ minimizes the sum of the individual variance components of the debiased GARS estimator, thereby ensuring component-wise small confidence intervals. Different notions of optimality are possible and have their origin in the literature on optimal experimental design (Atkinson et al., 2007). For example, we could minimize the determinant of $\Sigma(\pi)$ instead of the trace, leading to a smaller *volume of the confidence ellipsoid* from Eq. (17). In the following, we restrict ourselves to A-optimality as defined above, but we provide extensions to other optimality definitions in Appendix H.

**Deriving optimal labeling policies.** The following result explicitly characterizes A-optimal labeling functions.

**Theorem 6.2** (A-optimal labeling policy). *Assume* $\Pi_{\alpha,\beta} \neq \emptyset$ *(e.g.* $\alpha \sum c_{ij} \leq \beta$*) and let* $V_{jk}(\mu(x)) = \operatorname{Var}(Y_{jk} \mid X = x) = \operatorname{Diag}(\mu_{jk}(x)) - \mu_{jk}(x)\mu_{jk}(x)^\top \in \mathbb{R}^{C \times C}$.

*Under Assumption B.1 (Appendix), any A-optimal policy satisfies, for almost all* $x$ *and all* $j \neq k$,

$$
\pi_{jk}^*(x) = \operatorname{clip}_{[\alpha,1]} \sqrt{\frac{\operatorname{tr}\Big( J_{jk}(\mu(x))\, V_{jk}(\mu(x))\, J_{jk}(\mu(x))^\top \Big)}{\lambda_A\, c_{jk}}},
$$
(19)

*where* $\operatorname{clip}_{[\alpha,1]}(x) = \min\{1, \max\{\alpha, x\}\}$ *and where* $\lambda_A \geq 0$ *is chosen so that* $\mathbb{E}\big[\sum_{j \neq k} c_{jk} \pi_{jk}^*(X)\big] = \beta$.

*Proof.* See Appendix B. □

**Interpretation.** The A-optimal policy allocates more probability to pairs $(j, k)$ that are (i) intrinsically informative, meaning that $V_{jk}(\mu(x)) = \operatorname{Var}(Y_{jk} \mid X = x)$ is large (i.e., where humans tend to disagree more on their preferences); and (ii) influential for the GARS $\theta$, which is measured by the Jacobian $J_{jk}$. Additionally, the costs $c_{jk}$ down-weight expensive pairs via the denominator, and $\lambda_A$ scales the probabilities to satisfy the budget constraint.

**Implementation.** In practice, implementing the A-optimal labeling policy requires an estimate of the preference probabilities $\mu$. This can be obtained either (i) from historical preference data by fitting $\mu_{jkc}(x) = \mathbb{P}(Y_{jkc} = 1 \mid X = x, S_{jk} = 1)$, or (ii) directly from an external judge such as an LLM-as-a-judge or auto-rater that outputs estimated category probabilities $\hat{\mu}_{jk}(x)$.

Given $\hat{\mu}$, we then plug it into the closed-form rule in Theorem 6.2 to obtain $\hat{\pi}_{jk}^*(x)$. Since the left-hand side is monotone in $\lambda_A$, we estimate $\lambda_A$ efficiently via a one-dimensional binary search using the empirical average $\frac{1}{n} \sum_{i=1}^n \sum_{j \neq k} c_{jk} \hat{\pi}_{jk}^*(x_i)$. We then acquire new preference labels by *independent Bernoulli sampling*: for each context $x$ and each pair $(j, k)$, we draw $S_{jk} \sim \operatorname{Bernoulli}(\hat{\pi}_{jk}^*(x))$ and query a human label only if $S_{jk} = 1$. Optionally, to encourage exploration or improve robustness to misspecification of $\hat{\mu}$, one may mix the resulting policy with a baseline random policy, e.g., $\pi_{\mathrm{mix}}(x) = (1 - \varepsilon)\hat{\pi}^*(x) + \varepsilon\pi_{\mathrm{rand}}(x)$ for some small $\varepsilon \in (0, 1)$, while still enforcing the lower bound $\alpha$ and the budget constraint.

## 7. Experiments

**Implementation details.** We estimate nuisance functions using LightGBM, trained with standard gradient-boosted tree objectives for classification. As described in Sec. 5, we apply cross-fitting and train nuisance models held-out folds for predictions. The LightGBM hyperparameters are tuned via 3-fold cross-validation within the training folds (see Appendix K for details). We compute valid 95% simultaneous confidence intervals (CIs) for all ranking scores using multiple-testing adjustments (see Appendix I).

**Data.** As standard in the literature on debiased machine learning, we use various synthetic datasets for evaluating our proposed methodology (Chernozhukov et al., 2018). This is because synthetic data allows for comparison with ground-truth ranking scores, which are not available on real-world data. Nevertheless, we also apply our methodology to real-world datasets, including public preference data from Chatbot Arena and MT-Bench (Zheng et al., 2023). However, we emphasize that the score of our experiments is the evaluation of our methodology, not to provide a state-off-the-art leaderboard. We refer to Appendix L for details on synthetic data and to Appendix M for details on real-world data.

**Baselines.** We follow common benchmarking practices in debiased machine learning (Nie & Wager, 2021; Kennedy, 2023) and compare debiased estimators from Theorem 5.1 with plug-in estimators from Eq. (11) using the *same nuisance estimators* $\hat{\eta}$. We refrain from comparing different ML models for $\eta$ (e.g., different model architectures) as our primary goal is to show the advantages of model-agnostic debiased estimation.

### 7.1. Synthetic data

• **Effectiveness of debiasing.** Experimental setup. We compare the debiased GARS estimators for Borda scores, BT-scores, and RC-scores with their corresponding plug-in versions. For this, we generate synthetic preference data with varying sample sizes, $m = 3$ categories (win, loss, tie), and 5 synthetic covariates $X$. Results. The results are in Table 1. Across all GARS estimands and sampling regimes, debiased estimators significantly outperform their plug-in counterparts. Furthermore, debiased estimators achieve coverage close to the target of 95%, while the coverage of the plug-in estimator is invalid. *This confirms our main results from Theorem 5.1 (statistical efficiency and asymptotic*

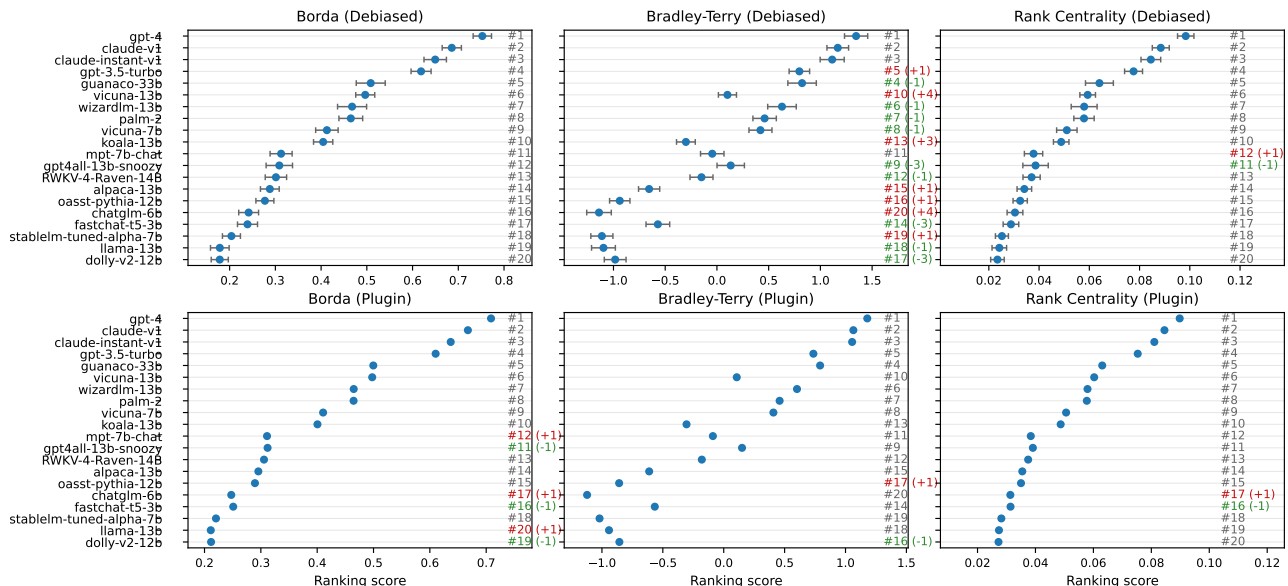

*Figure 3.* **Results for Chatbot Arena preference data.** Shown: estimated ranking scores for different GARS functionals (Borda, BT, and rank centrality) and estimators (debiased and plug-in). Changes in ranking are indicated in red and green as compared to the baseline ranking from Zheng et al. (2023). We report 95% simultaneous confidence intervals, which attain near-zero width for plug-in estimators.

*normality).*

*Table 1.* **Experimental results for debiasing.** Shown: Estimation error an CI coverage (mean ± 95% CIs over 100 runs). Best in bold. Valid coverage (within CI) in green, otherwise in red.

| GARS | Estimator | $n = 1000$ | | $n = 2000$ | | $n = 3000$ | |
|---|---|---|---|---|---|---|---|
| | | Error | Coverage | Error | Coverage | Error | Coverage |
| Borda | Plug-in | $0.38 \pm 0.08$ | $0.17 \pm 0.09$ | $0.16 \pm 0.04$ | $0.20 \pm 0.09$ | $0.10 \pm 0.02$ | $0.15 \pm 0.08$ |
| | Debiased | $\mathbf{0.15 \pm 0.03}$ | $\mathbf{0.94 \pm 0.06}$ | $\mathbf{0.08 \pm 0.02}$ | $\mathbf{0.95 \pm 0.06}$ | $\mathbf{0.05 \pm 0.01}$ | $\mathbf{0.97 \pm 0.05}$ |
| BT | Plug-in | $0.62 \pm 0.13$ | $0.09 \pm 0.07$ | $0.30 \pm 0.07$ | $0.07 \pm 0.07$ | $0.22 \pm 0.04$ | $0.05 \pm 0.06$ |
| | Debiased | $\mathbf{0.25 \pm 0.05}$ | $\mathbf{0.90 \pm 0.07}$ | $\mathbf{0.12 \pm 0.02}$ | $\mathbf{0.85 \pm 0.08}$ | $\mathbf{0.08 \pm 0.01}$ | $\mathbf{0.90 \pm 0.07}$ |
| RC | Plug-in | $0.52 \pm 0.12$ | $0.12 \pm 0.08$ | $0.26 \pm 0.06$ | $0.06 \pm 0.06$ | $0.18 \pm 0.04$ | $0.05 \pm 0.06$ |
| | Debiased | $\mathbf{0.27 \pm 0.06}$ | $\mathbf{0.91 \pm 0.07}$ | $\mathbf{0.13 \pm 0.02}$ | $\mathbf{0.95 \pm 0.06}$ | $\mathbf{0.09 \pm 0.02}$ | $\mathbf{0.95 \pm 0.06}$ |

• **Using external judges.** Experimental setup. We evaluate the effectiveness of augmenting debiased estimators with external judges of different quality. To simulate judges, we add noise of different strengths to the ground-truth nuisance $\mu$ and use judge-as-features as described in Sec. 5. Results. Figure 4a shows the results across different judge noise levels. *All debiased estimators achieve lower estimation error when incorporating higher quality judges, thus confirming the effectiveness of our method to leverage external judges.*

• **Optimal data acquisition.** Experimental setup. We compare the A-optimal data-collection policy from Theorem 6.2 with a uniform baseline that collects preference data at random. For this, we label $n = 1500$ contexts $X$ two times with both policies using a budget of $\beta = 2000$ each for three different GARS types. We then compare the estimation error of the corresponding debiased estimators. Results. The results are shown in Table 2. *The A-optimal policy outperforms random collection across all GARS types, thus confirming our result from Theorem 6.2.*

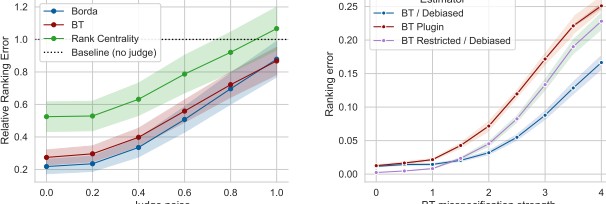

*(a)* Results for synthetic judges    *(b)* Results for BT misspecification

*Figure 4.* **Synthetic experimental results.** (a) Relative ranking error normalized by the no-judge baseline (mean and 95% CIs over $n = 100$ runs). (b) BT-projection estimation error (mean and 95% CIs over 30 runs).

*Table 2.* **Experimental results for preference data acquisition.** Shown: Ranking MSE (Mean and 95% confidence intervals over 50 runs $(\cdot 10^2)$) of debiased GARS estimators using the data from two labeling policies at budget $n = 2000$. Best in bold.

| Policy | Borda | BT | Rank Centrality |
|---|---|---|---|
| A-optimal policy | $\mathbf{0.130 \pm 0.031}$ | $\mathbf{2.861 \pm 0.670}$ | $\mathbf{0.017 \pm 0.004}$ |
| Random policy | $0.141 \pm 0.030$ | $2.974 \pm 0.585$ | $0.020 \pm 0.005$ |

• **BT model misspecification.** Experimental setup. Finally, we evaluate the effect of misspecification of the BT model. We create synthetic datasets with binary preferences deviating from a true BT model by a parameter $\gamma$, where $\gamma = 0$ equals a true BT model. We then compare three estimators: (i) the debiased BT projection estimator from Sec.4.1, (ii) the corresponding plug-in estimator, and (iii) a restricted debiased estimator which assumes that a BT model holds (see Appendix F for details). Results. Fig.4b shows that

the restricted debiased estimator achieves the best performance under a correct BT model specification due to a lower efficiency bound in the setting. However, under BT misspecification, the debiased projection estimator outperforms both the plug-in and the misspecified debiased estimator. *This highlights the robustness of the debiased projection estimator to BT model misspecifications.*

### 7.2. Real-world data

**Experimental setup.** We demonstrate DMLRANK using a public, real-world preference dataset with conversations from Chatbot Arena (Zheng et al., 2023). The dataset includes $n = 32980$ different prompts, $K = 20$ different models a categorical preference outcome in {*first model wins, second model wins, tie, tie (both bad)*}. Context features $X$ combine a toxicity probability (from a RoBERTa-based tag) with a model-agnostic low-rank 100-dimensional representation of the prompt, obtained via TF-IDF followed by a truncated SVD. We report estimated GARS scores and 95% simultaneous confidence intervals for Borda, BT, and Rank Centrality using both plug-in and debiased estimators.

**Results.** The results are shown in Fig. 3, with the baseline ranking of models taken from (Zheng et al., 2023) (note that this is *not* a ground-truth ranking). We make the following observations: (i) all obtained rankings are largely consistent with the baseline ranking, thus indicating a reasonable base performance. (ii) BT deviates further from the baseline than Borda and RC, thus highlighting that different GARS types can yield different rankings even on the same data. (iii) While the point ranking scores of the plug-in estimators do not deviate too much from their debiased counterparts, their confidence intervals collapse and are most likely invalid. In summary, the results showcase the applicability and flexibility of the proposed GARS framework for real-world preference data.

## 8. Discussion

We proposed a nonparametric statistical framework for preference-based LLM evaluation that targets ranking functionals directly from contextual preference data. By formulating evaluation through generalized average ranking scores, our approach unifies several common ranking procedures while allowing flexible nonparametric estimation. We developed debiased estimators with valid uncertainty quantification, showed how auxiliary judges can be incorporated without treating them as ground truth, and studied optimal data collection policies for improving the efficiency of leaderboard estimation.

**Limitations and future work.** Our work does not capture all aspects of evaluating modern AI systems. In particular, many evaluations involve complex systems with tools,

memory, multi-agent interactions, long-horizon tasks, or deployment-specific feedback loops. Extending the framework to such settings is an important direction for future work. Another key practical limitation is lack of overlap. When some model pairs, contexts, or annotator subpopulations have very small selection probabilities, estimators based on inverse propensity weighting can become unstable. Practical implementations may require additional stabilization, such as trimming, clipping, or propensity calibration as establied in causal inference literature. Finally, our results rely on missing-at-random assumptions for the observed preference labels. Violations can introduce bias that cannot be removed by debiasing alone. A promising direction is to adapt partial identification approaches to preference-based LLM evaluation, yielding uncertainty sets or bounds for ranking scores when missingness assumptions are weakened.

**Conclusion.** By combining flexible ranking estimands, debiased estimation, uncertainty quantification, auxiliary judges, and efficient data collection, the proposed framework offers methodology for more reliable and transparent leaderboards.

## Impact Statement

This paper presents work whose goal is to advance the field of machine learning. There are many potential societal consequences of our work, none of which we feel must be specifically highlighted here.

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

# A. Extended related work on experimental design

Optimal experimental design has a long history (Atkinson et al., 2007), including classical variance-minimizing allocation rules such as Neyman allocation (Neyman, 1934) and their extensions to semiparametric causal inference and semi-supervised learning settings (Cook et al., 2024; Oprescu et al., 2025; Zrnic & Candès, 2024). In the context of LLM evaluation, Angelopoulos et al. (2025) study cost-optimal active evaluation, but not preference-based ranking targets. There is also a line of work on actively selecting pairwise comparisons for ranking (Jamieson & Nowak, 2011) and on optimal design under parametric BT/BTL models (Graßhoff & Schwabe, 2008; Guo et al., 2018). Finally, dueling bandits (Yue & Joachims, 2009; Dudík et al., 2015; Mukherjee et al., 2024) study interactive comparison feedback but typically target regret minimization rather than efficient statistical inference for a fixed ranking estimand. In contrast, our design problem is *efficient influence function (EIF)-driven*: we derive labeling policies that minimize the asymptotic variance of semiparametrically efficient estimators for general (nonparametric) ranking functionals.

# B. Theoretical results

## B.1. Setup

**Identifiability assumptions.** We make the following assumptions (Rubin, 1974): (i) Positivity: $\pi_{jk}(x) > 0$ whenever $\mathbb{P}(X = x) > 0$ (every pair of items has a positive selection probability), and (ii) Missing at random: $Y \perp\!\!\!\perp S \mid X$ (the preference label does not affect missingness). Note that (ii) implies that $\mu_{jkc}(x) = \mathbb{P}(Y_{jkc} = 1 \mid X = x, S_{jk} = 1)$ which can be estimated from the observed data $\mathcal{D}$.

Assumptions (i) and (ii) are standard in the literature on missing data and causal inference (Rubin, 1974). If they fail, the population ranking target may no longer be identifiable from the observed data alone. In practice, this means observed labels are then systematically biased, and *no method can guarantee unbiased estimation* without adding further assumptions or information. Note that (i) can be ensured by design (e.g., via randomized exploration to ensure all pairs have non-negligible selection probability).

**DML assumptions.** We work under the standard conditions used for one-step / DML estimators with cross-fitting (van der Vaart, 1998; Chernozhukov et al., 2018):

(A1) **Sampling.** The observations $o_i = (x_i, a_i, \widetilde{y}_i)$ are i.i.d. draws from $\mathbb{P}$.

(A2) **Smoothness of the target map.** The function $F : [0, 1]^{K \times K \times C} \to \mathbb{R}^d$ is (Fréchet) differentiable on a neighborhood of $\{\mu(x) : x \in \mathcal{X}\}$ with Jacobian blocks $J_{jk}(\mu) = \nabla_{\mu_{jk}} F(\mu) \in \mathbb{R}^{d \times C}$. Moreover, the Jacobian is locally Lipschitz: there is $L < \infty$ such that for all $\mu, \mu'$ in this neighborhood,

$$\|J_{jk}(\mu) - J_{jk}(\mu')\| \leq L\|\mu - \mu'\|, \qquad \forall j \neq k,$$

and $\sup_x \|J_{jk}(\mu(x))\| < \infty$ for all $j \neq k$.

(A3) **Moments.** $\mathbb{E}[\|\phi(O, \eta)\|^2] < \infty$.

(A4) **Cross-fitted nuisance estimation.** The nuisance estimators $\hat{\eta} = (\hat{\mu}, \hat{\pi})$ are obtained by cross-fitting on $V \geq 2$ folds as described in Sec.5. In addition, the nuisance estimators satisfy the usual product-rate conditions

$$\sum_{j \neq k} \|\hat{\mu}_{jk} - \mu_{jk}\|_{L^2(P)}^2 = o_p(n^{-1/2}), \qquad \sum_{j \neq k} \|\hat{\mu}_{jk} - \mu_{jk}\|_{L^2(P)} \|\hat{\pi}_{jk} - \pi_{jk}\|_{L^2(P)} = o_p(n^{-1/2}),$$

which is implied, for instance, by $\|\hat{\mu} - \mu\|_{L^2} = o_p(n^{-1/4})$ and $\|\hat{\pi} - \pi\|_{L^2} = o_p(n^{-1/4})$ in fixed $K$ settings. Furthermore, we assume that $\inf_{x, j \neq k} \hat{\pi}_{jk}(x) \geq \underline{\pi}/2$ with probability tending to one (e.g., via truncation).

**Optimal data acquisition.** For deriving the A-optimal policy in Theorem 6.2, we impose the following assumption.

**Assumption B.1** (Independent selection)**.** *For all $x \in \mathcal{X}$ and $s \in \{0, 1\}^{K \times K}$, it holds*

$$\mathbb{P}(S = s \mid X = x) = \prod_{j \neq k} \pi_{jk}(x)^{s_{jk}} (1 - \pi_{jk}(x))^{1 - s_{jk}}.$$

Assumption B.1 ensures that labeling a preference for pair $(j, k)$ does not affect the labeling for a different pair $(\ell, m)$. This is satisfied in many practical applications, for example, in adaptive experimentation, where the data is already generated by a previous labeling policy that we control. However, we emphasize that we only rely on Assumption B.1 for *optimality* of our labeling policy. Even under violations of Assumption B.1, our labeling policy may still yield high-quality labels, especially when compared to a policy that collects labels at random.

## B.2. Proof of Theorem 5.1

*Proof.* We prove (i) the efficient influence function (EIF) in Eq. ((13)), and (ii) asymptotic linearity, normality, and efficiency of the one-step estimator from Eq. ((14)) under standard regularity conditions.

We start by defining the parameter map

$$\theta(\mathbb{P}) = \mathbb{E}_{\mathbb{P}}[F(\mu_{\mathbb{P}}(X))] \in \mathbb{R}^d,$$

where $\mu_{\mathbb{P}}$ denotes the regression function under $\mathbb{P}$.

**Step 1: Derivation of the EIF.** Consider a regular parametric submodel $\{\mathbb{P}_\varepsilon : \varepsilon \in (-\delta, \delta)\}$ through $\mathbb{P}_0 = \mathbb{P}$ with score $s(O) = \partial_\varepsilon \log \mathbb{P}_\varepsilon(O)|_{\varepsilon=0}$. Under missing at random, the observed-data likelihood factorizes as

$$\mathbb{P}(o) = \mathbb{P}(x)\,\mathbb{P}(s \mid x)\,\mathbb{P}(\widetilde{y} \mid x, s),$$

and the corresponding score decomposes as

$$s(O) = s_X(X) + s_{S|X}(S \mid X) + s_{\widetilde{Y}|X,S}(\widetilde{Y} \mid X, S),$$

with $\mathbb{E}[s_X(X)] = 0$, $\mathbb{E}[s_{S|X}(S \mid X) \mid X] = 0$, and $\mathbb{E}[s_{\widetilde{Y}|X,S}(\widetilde{Y} \mid X, S) \mid X, S] = 0$.

We compute the pathwise derivative of $\theta(\mathbb{P}_\varepsilon)$ at $\varepsilon = 0$. Write $\mu_\varepsilon$ for the regression under $\mathbb{P}_\varepsilon$ and $\theta_\varepsilon = \theta(\mathbb{P}_\varepsilon)$. By the product rule, we have

$$\begin{aligned}
\partial_\varepsilon \theta_\varepsilon|_0 &= \partial_\varepsilon \mathbb{E}_\varepsilon\big[F\big(\mu_\varepsilon(X)\big)\big]\big|_0 \\
&= \mathbb{E}\big[\big(F(\mu(X)) - \theta\big)\,s_X(X)\big] \;+\; \mathbb{E}\big[\partial_\varepsilon F\big(\mu_\varepsilon(X)\big)\big|_0\big].
\end{aligned}$$

Since $F$ is differentiable, the chain rule yields

$$\partial_\varepsilon F\big(\mu_\varepsilon(X)\big)\big|_0 = \sum_{j \neq k} J_{jk}\big(\mu(X)\big)\,\partial_\varepsilon \mu_{jk,\varepsilon}(X)|_0.$$

*Key identity (regression under MAR).* Fix $(j,k)$ and any component $c \in \{1, \ldots, C\}$. Because $Y \perp S \mid X$ and $\widetilde{Y}_{jk} = S_{jk} Y_{jk}$, we have, for all $x$, that

$$\mu_{jk,c}(x) = \mathbb{E}[Y_{jk,c} \mid X = x] = \mathbb{E}[Y_{jk,c} \mid X = x, S_{jk} = 1].$$

For a regular submodel that perturbs the conditional law of labels (and/or $\mathbb{P}(x)$), standard likelihood calculus gives

$$\partial_\varepsilon \mu_{jk,c,\varepsilon}(x)|_0 = \mathbb{E}\left[\frac{S_{jk}}{\pi_{jk}(X)}\,(Y_{jk,c} - \mu_{jk,c}(X))\,s(O)\,\Big|\,X = x\right].$$

This follows from (i) $\mathbb{E}[S_{jk}/\pi_{jk}(X) \mid X] = 1$, (ii) $\mathbb{E}[Y_{jk,c} - \mu_{jk,c}(X) \mid X] = 0$, and (iii) the fact that $s_{\widetilde{Y}|X,S}$ spans mean-zero perturbations of the conditional law of the observed labels given $(X, S)$; the factor $S_{jk}/\pi_{jk}(X)$ is the usual coarsening adjustment under MAR/positivity.

Multiplying by $J_{jk}(\mu(X))$ and taking expectation over $X$ yields

$$\mathbb{E}\left[\sum_{j \neq k} J_{jk}\big(\mu(X)\big)\,\partial_\varepsilon \mu_{jk,\varepsilon}(X)|_0\right] = \mathbb{E}\left[\sum_{j \neq k} \frac{S_{jk}}{\pi_{jk}(X)}\,J_{jk}\big(\mu(X)\big)\big(Y_{jk} - \mu_{jk}(X)\big)\,s(O)\right],$$

where $Y_{jk}, \mu_{jk}(X) \in \mathbb{R}^C$ and the product is matrix–vector multiplication.

Putting the pieces together, we obtain

$$\partial_\varepsilon \theta_\varepsilon|_0 = \mathbb{E}\left[\left\{F\big(\mu(X)\big) - \theta + \sum_{j \neq k} \frac{S_{jk}}{\pi_{jk}(X)}\,J_{jk}\big(\mu(X)\big)\big(Y_{jk} - \mu_{jk}(X)\big)\right\}\,s(O)\right].$$

Therefore, the candidate influence function

$$\phi(O, \eta) = F\big(\mu(X)\big) - \theta + \sum_{j \neq k} \frac{S_{jk}}{\pi_{jk}(X)}\,J_{jk}\big(\mu(X)\big)\big(Y_{jk} - \mu_{jk}(X)\big)$$

satisfies the defining property of the (canonical) gradient: for every regular submodel with score $s$, the pathwise derivative equals $\mathbb{E}[\phi s]$.

It remains to verify $\phi$ is efficient. Because $\theta(\mathbb{P})$ does not depend on $\mathbb{P}(s \mid x)$, the nuisance tangent space contains $\{h(X, S) : \mathbb{E}[h \mid X] = 0\}$. We have $\mathbb{E}[\phi(O, \eta) \mid X, S] = F(\mu(X)) - \theta$ since $\mathbb{E}[Y_{jk} - \mu_{jk}(X) \mid X, S_{jk} = 1] = 0$ and the summand vanishes when $S_{jk} = 0$. Hence, $\mathbb{E}[\phi\,h(X, S)] = 0$ for all such $h$, i.e., $\phi$ is orthogonal to the missingness tangent space. Thus $\phi$ is the projection of the gradient onto the orthocomplement of the nuisance tangent space, i.e., the EIF. This proves Eq. (13).

**Step 2: One-step estimator, asymptotic linearity and normality.** Define the one-step estimator

$$\hat{\theta}_{\mathrm{EIF}} = \frac{1}{n}\sum_{i=1}^{n}\left[F(\hat{\mu}(x_i)) + \sum_{j\neq k}\frac{s_{i,jk}}{\hat{\pi}_{jk}(x_i)}J_{jk}(\hat{\mu}(x_i))\big(y_{i,jk} - \hat{\mu}_{jk}(x_i)\big)\right],$$

where $(\hat{\mu}, \hat{\pi})$ are cross-fitted nuisance estimates.

Let $P_n$ denote the empirical measure and $P$ the true distribution. Write $\phi(O, \eta)$ for the EIF and $\phi(O, \hat{\eta})$ for the same expression with $(\mu, \pi)$ replaced by $(\hat{\mu}, \hat{\pi})$ and $\theta$ replaced by $\hat{\theta}_{\mathrm{EIF}}$ where appropriate. A standard one-step expansion yields

$$\hat{\theta}_{\mathrm{EIF}} - \theta = (\mathbb{P}_n - \mathbb{P})\phi(O, \eta) \;+\; \underbrace{\mathbb{P}\{\phi(O, \hat{\eta}) - \phi(O, \eta)\}}_{=:R_n} \;+\; \underbrace{(\mathbb{P}_n - \mathbb{P})\{\phi(O, \hat{\eta}) - \phi(O, \eta)\}}_{=:r_n}.$$

Cross-fitting implies $r_n = o_p(n^{-1/2})$ under mild moment conditions because, on each fold, $\phi(O_i, \hat{\eta})$ is evaluated using nuisance estimates trained on independent data, allowing a conditional central limit theorem (CLT)/conditional law of large numbers (LLN) argument (see, e.g., standard cross-fitting results for one-step/DML estimators).

**Control of the remainder $R_n$.** Recall $R_n := \mathbb{P}\{\phi(O, \hat{\eta}) - \phi(O, \eta)\} = \mathbb{P}\phi(O, \hat{\eta})$ since $\mathbb{P}\phi(O, \eta) = 0$. Using iterated expectations and MAR, for each $(j, k)$, we have

$$\mathbb{E}\left[\frac{S_{jk}}{\hat{\pi}_{jk}(X)}\big(Y_{jk} - \hat{\mu}_{jk}(X)\big)\,\Big|\,X\right] = \frac{\pi_{jk}(X)}{\hat{\pi}_{jk}(X)}\big(\mu_{jk}(X) - \hat{\mu}_{jk}(X)\big),$$

and therefore

$$R_n = \mathbb{E}\Big[F(\hat{\mu}(X)) - F(\mu(X))\Big] + \sum_{j\neq k}\mathbb{E}\Big[\frac{\pi_{jk}(X)}{\hat{\pi}_{jk}(X)}J_{jk}(\hat{\mu}(X))\big(\mu_{jk}(X) - \hat{\mu}_{jk}(X)\big)\Big].$$

Add and subtract the linear term $\sum_{j\neq k}J_{jk}(\mu(X))(\hat{\mu}_{jk}(X) - \mu_{jk}(X))$ to obtain a decomposition $R_n = A_n + B_n + C_n$, where $A_n$ is the second-order Taylor remainder of $F$ around $\mu$, $B_n$ collects the error from replacing $J_{jk}(\mu(X))$ by $J_{jk}(\hat{\mu}(X))$, and $C_n$ collects the error from replacing $\pi_{jk}(X)$ by $\hat{\pi}_{jk}(X)$ in the inverse-weight factor. By the locally Lipschitz property of the Jacobian (Taylor theorem in integral form), there exists a constant $L < \infty$ such that

$$\Big\|F(\hat{\mu}(X)) - F(\mu(X)) - \sum_{j\neq k}J_{jk}(\mu(X))(\hat{\mu}_{jk}(X) - \mu_{jk}(X))\Big\| \;\leq\; L\|\hat{\mu}(X) - \mu(X)\|^2,$$

and similarly $\|J_{jk}(\hat{\mu}(X)) - J_{jk}(\mu(X))\| \leq L\|\hat{\mu}(X) - \mu(X)\|$. Moreover, under positivity and truncation $\inf_{x,j\neq k}\hat{\pi}_{jk}(x) \geq \underline{\pi}/2$ with high probability, the ratios $\pi_{jk}(X)/\hat{\pi}_{jk}(X)$ are uniformly bounded and $\big|\pi_{jk}(X)/\hat{\pi}_{jk}(X) - 1\big| \lesssim |\hat{\pi}_{jk}(X) - \pi_{jk}(X)|$. Combining these bounds and applying Cauchy–Schwarz to the terms involving $\hat{\pi} - \pi$ yields the generic estimate

$$\|R_n\| \;\lesssim\; \sum_{j\neq k}\Big(\|\hat{\mu}_{jk} - \mu_{jk}\|_{L^2(P)}^2 + \|\hat{\mu}_{jk} - \mu_{jk}\|_{L^2(P)}\|\hat{\pi}_{jk} - \pi_{jk}\|_{L^2(P)}\Big),$$

which is $o_p(n^{-1/2})$ under the rate conditions in Assumption (A5).

Consequently,

$$\sqrt{n}(\hat{\theta}_{\mathrm{EIF}} - \theta) = \frac{1}{\sqrt{n}}\sum_{i=1}^{n}\phi(O_i, \eta) + o_p(1).$$

By the multivariate central limit theorem and $\mathbb{E}[\phi(O, \eta)] = 0$, we obtain

$$\sqrt{n}(\hat{\theta}_{\mathrm{EIF}} - \theta) \;\rightsquigarrow\; \mathcal{N}_d\big(0, \Sigma\big), \qquad \Sigma = \mathbb{E}\big[\phi(O, \eta)\phi(O, \eta)^{\top}\big].$$

**Step 3: Efficiency.** Since $\phi(O, \eta)$ is the EIF (canonical gradient), $\Sigma$ is the semiparametric efficiency bound. The asymptotic linear representation above therefore shows that $\hat{\theta}_{\mathrm{EIF}}$ achieves this bound, i.e., it is asymptotically efficient.

This completes the proof. $\qquad\square$

## B.3. Proof of Theorem 6.2

*Proof.* Fix $\pi \in \Pi_{\alpha,\beta}$ and recall from Theorem 5.1 that the EIF is

$$\phi(O, \eta, \theta) = F(\mu(X)) - \theta + \sum_{j \neq k} \frac{S_{jk}}{\pi_{jk}(X)} J_{jk}(\mu(X))\big(Y_{jk} - \mu_{jk}(X)\big).$$

Write

$$A(X) := F(\mu(X)) - \theta, \qquad B_\pi(O) := \sum_{j \neq k} \frac{S_{jk}}{\pi_{jk}(X)} J_{jk}(\mu(X))\big(Y_{jk} - \mu_{jk}(X)\big),$$

so that $\phi(O, \eta, \theta) = A(X) + B_\pi(O)$. Recall that the efficiency bound equals

$$\Sigma(\pi) = \mathbb{E}\big[\phi(O, \eta, \theta)\phi(O, \eta, \theta)^\top\big].$$

which can be expanded to

$$\Sigma(\pi) = \mathbb{E}[A(X)A(X)^\top] + \mathbb{E}[B_\pi(O)B_\pi(O)^\top] + 2\,\mathbb{E}[A(X)B_\pi(O)^\top].$$

By the missing at random assumption ($Y \perp\!\!\!\perp S \mid X$) we have, for each $j \neq k$,

$$\mathbb{E}\big[J_{jk}(\mu(X))\big(Y_{jk} - \mu_{jk}(X)\big) \mid X\big] = J_{jk}(\mu(X))\,\mathbb{E}[Y_{jk} - \mu_{jk}(X) \mid X] = 0,$$

hence $\mathbb{E}[B_\pi(O) \mid X] = 0$ and thus

$$\mathbb{E}[A(X)B_\pi(O)^\top] = \mathbb{E}\big[A(X)\mathbb{E}[B_\pi(O)^\top \mid X]\big] = 0.$$

Therefore, the $\pi$-dependence of $\Sigma(\pi)$ is entirely through $\mathbb{E}[B_\pi(O)B_\pi(O)^\top]$.

**Step 1: Reducing $\mathrm{tr}(\Sigma(\pi))$ to a separable objective.** Condition on $X = x$ and define

$$U_{jk}(x) := J_{jk}(\mu(x))\big(Y_{jk} - \mu_{jk}(x)\big) \in \mathbb{R}^d.$$

Then $B_\pi(O) \mid (X = x) = \sum_{j \neq k} \frac{S_{jk}}{\pi_{jk}(x)} U_{jk}(x)$, so

$$\mathbb{E}[B_\pi(O)B_\pi(O)^\top \mid X = x] = \sum_{j \neq k} \sum_{\ell \neq m} \mathbb{E}\Big[\frac{S_{jk}S_{\ell m}}{\pi_{jk}(x)\pi_{\ell m}(x)} \Big| X = x\Big] \mathbb{E}[U_{jk}(x)U_{\ell m}(x)^\top \mid X = x],$$

where we used $Y \perp\!\!\!\perp S \mid X$ to separate selection and label moments. Under Assumption B.1,

$$\mathbb{E}\Big[\frac{S_{jk}S_{\ell m}}{\pi_{jk}(x)\pi_{\ell m}(x)} \Big| X = x\Big] = \begin{cases} \frac{1}{\pi_{jk}(x)}, & (j,k) = (\ell, m), \\ 1, & (j,k) \neq (\ell, m). \end{cases}$$

Hence

$$\mathbb{E}[B_\pi(O)B_\pi(O)^\top \mid X = x] = \sum_{j \neq k} \frac{1}{\pi_{jk}(x)} \mathbb{E}[U_{jk}(x)U_{jk}(x)^\top \mid X = x] + \sum_{\substack{(j,k) \neq (\ell,m) \\ j \neq k,\ \ell \neq m}} \mathbb{E}[U_{jk}(x)U_{\ell m}(x)^\top \mid X = x].$$

The second (off-diagonal) term does not involve $\pi$, so taking traces and then expectations over $X$ yields

$$\mathrm{tr}(\Sigma(\pi)) = \mathrm{const} + \mathbb{E}\left[\sum_{j \neq k} \frac{1}{\pi_{jk}(X)} \mathrm{tr}\Big(\mathbb{E}[U_{jk}(X)U_{jk}(X)^\top \mid X]\Big)\right],$$

where "const" collects all $\pi$-free terms.

Next, we compute the conditional variance. Since

$$\mathrm{Var}(Y_{jk} \mid X = x) = V_{jk}(\mu(x)) = \mathrm{Diag}(\mu_{jk}(x)) - \mu_{jk}(x)\mu_{jk}(x)^\top,$$

we have

$$\mathbb{E}\big[U_{jk}(x)U_{jk}(x)^{\top} \mid X = x\big] = J_{jk}(\mu(x))\,\mathbb{E}\big[(Y_{jk} - \mu_{jk}(x))(Y_{jk} - \mu_{jk}(x))^{\top} \mid X = x\big]\,J_{jk}(\mu(x))^{\top}$$
$$= J_{jk}(\mu(x))\,V_{jk}(\mu(x))\,J_{jk}(\mu(x))^{\top}.$$

Define the nonnegative scalar weight

$$a_{jk}(x) := \mathrm{tr}\Big(J_{jk}(\mu(x))\,V_{jk}(\mu(x))\,J_{jk}(\mu(x))^{\top}\Big) \geq 0.$$

Then minimizing $\mathrm{tr}(\Sigma(\pi))$ over $\Pi_{\alpha,\beta}$ is equivalent to solving

$$\min_{\pi:\,\mathcal{X}\to[\alpha,1]^{K\times K}}\ \mathbb{E}\left[\sum_{j\neq k}\frac{a_{jk}(X)}{\pi_{jk}(X)}\right] \quad \text{s.t.} \quad \mathbb{E}\left[\sum_{j\neq k}c_{jk}\pi_{jk}(X)\right] \leq \beta. \tag{20}$$

**Step 2: Lagrangian and pointwise minimization.** Introduce a multiplier $\lambda_A \geq 0$ for the budget constraint in Eq. ((20)). The Lagrangian is

$$\mathcal{L}(\pi, \lambda_A) = \mathbb{E}\left[\sum_{j\neq k}\left(\frac{a_{jk}(X)}{\pi_{jk}(X)} + \lambda_A c_{jk}\pi_{jk}(X)\right)\right] - \lambda_A\beta,$$

with box constraints $\pi_{jk}(x) \in [\alpha, 1]$ imposed explicitly. For fixed $\lambda_A$, the integrand is separable across $(x, j, k)$, so, for $x$ almost everywhere (a.e.) and each $(j, k)$, we minimize

$$g_{jk,x}(\pi) := \frac{a_{jk}(x)}{\pi} + \lambda_A c_{jk}\pi \quad \text{over} \quad \pi \in [\alpha, 1].$$

On $(0, \infty)$, $g'_{jk,x}(\pi) = -a_{jk}(x)/\pi^2 + \lambda_A c_{jk}$, so the unique unconstrained minimizer satisfies

$$\pi = \sqrt{\frac{a_{jk}(x)}{\lambda_A c_{jk}}}.$$

Projecting onto $[\alpha, 1]$ yields

$$\pi^*_{jk}(x) = \mathrm{clip}_{[\alpha,1]}\sqrt{\frac{a_{jk}(x)}{\lambda_A c_{jk}}} = \mathrm{clip}_{[\alpha,1]}\sqrt{\frac{\mathrm{tr}\Big(J_{jk}(\mu(x))\,V_{jk}(\mu(x))\,J_{jk}(\mu(x))^{\top}\Big)}{\lambda_A\,c_{jk}}}.$$

**Step 3: Choosing $\lambda_A$ to satisfy the budget.** Define

$$G(\lambda) := \mathbb{E}\left[\sum_{j\neq k}c_{jk}\,\mathrm{clip}_{[\alpha,1]}\sqrt{\frac{a_{jk}(X)}{\lambda\,c_{jk}}}\right].$$

For each $(j, k)$ and a.e. $x$, the map $\lambda \mapsto \mathrm{clip}_{[\alpha,1]}\sqrt{a_{jk}(x)/(\lambda c_{jk})}$ is nonincreasing, hence $G(\lambda)$ is nonincreasing in $\lambda$. Moreover, $\lim_{\lambda\to\infty}G(\lambda) = \mathbb{E}[\sum_{j\neq k}c_{jk}\alpha]$ and $\lim_{\lambda\downarrow 0}G(\lambda) = \mathbb{E}[\sum_{j\neq k}c_{jk}]$ due to clipping at 1. Since $\Pi_{\alpha,\beta} \neq \emptyset$ implies $\beta \geq \mathbb{E}[\sum_{j\neq k}c_{jk}\alpha]$, there exists $\lambda_A \geq 0$ such that $G(\lambda_A) = \beta$ whenever the constraint is binding; if the budget is slack, the minimizer is $\pi^*_{jk}(x) \equiv 1$ (corresponding to $\lambda_A = 0$). This proves the stated characterization of any A-optimal policy. $\quad\square$

### B.4. Robustness of plug-in A-optimal acquisition

The following result quantifies robustness of our data-collection policy w.r.t. nuisance estimation error.

**Theorem B.2.** *For any measurable $\nu : \mathcal{X} \to [0,1]^{K \times K \times C}$, define*

$$V_{jk}(\nu(x)) := Diag(\nu_{jk}(x)) - \nu_{jk}(x)\nu_{jk}(x)^\top,$$

*and*

$$a_{jk}(x;\nu) := tr\Big( J_{jk}(\nu(x)) \, V_{jk}(\nu(x)) \, J_{jk}(\nu(x))^\top \Big).$$

*Let*

$$Q(\pi;\nu) := \mathbb{E}\left[ \sum_{j \neq k} \frac{a_{jk}(X;\nu)}{\pi_{jk}(X)} \right], \qquad \pi \in \Pi_{\alpha,\beta}.$$

*Let $\pi^* \in \arg\min_{\pi \in \Pi_{\alpha,\beta}} Q(\pi;\mu)$ be the oracle A-optimal policy, and let*

$$\tilde{\pi} \in \arg\min_{\pi \in \Pi_{\alpha,\beta}} Q(\pi;\hat{\mu})$$

*be the plug-in A-optimal policy computed from an estimate $\hat{\mu}$.*

*For a fixed labeling policy $q \in \Pi_{\alpha,\beta}$, let $\hat{\theta}^{(q)}_{\mathrm{EIF}}$ denote the debiased estimator in Eq. (14) computed from data collected under $q$, with known realized propensities, i.e., with $\hat{\pi} = q$. Assume the conditions of Theorem 5.1 hold for every fixed $q \in \Pi_{\alpha,\beta}$, and moreover that*

$$R_n(q) := \mathbb{E}\left[ \left\| \hat{\theta}^{(q)}_{\mathrm{EIF}} - \theta \right\|_2^2 \right] = \frac{1}{n} tr\big(\Sigma(q)\big) + o(n^{-1}).$$

*Then*

$$R_n(\tilde{\pi}) - R_n(\pi^*) \;\leq\; \frac{2}{\alpha n} \mathbb{E}\left[ \sum_{j \neq k} \big| a_{jk}(X;\hat{\mu}) - a_{jk}(X;\mu) \big| \right] + o(n^{-1}).$$

*In particular, if there exists $L_a < \infty$ such that*

$$|a_{jk}(x;\nu) - a_{jk}(x;\nu')| \leq L_a \|\nu(x) - \nu'(x)\|_\infty \qquad \text{for all } x, \ j \neq k,$$

*then any uniform error bound $\|\hat{\mu} - \mu\|_\infty \leq \varepsilon$ implies*

$$R_n(\tilde{\pi}) - R_n(\pi^*) \;\leq\; \frac{2K(K-1)L_a}{\alpha n}\varepsilon + o(n^{-1}).$$

*Proof.* As shown in Step 1 of the proof of Theorem 6.2,

$$tr\big(\Sigma(\pi)\big) = \mathrm{const} + Q(\pi;\mu),$$

where the constant does not depend on $\pi$. Hence it is enough to bound

$$Q(\tilde{\pi};\mu) - Q(\pi^*;\mu).$$

Add and subtract the plug-in objective:

$$Q(\tilde{\pi};\mu) - Q(\pi^*;\mu) = \big(Q(\tilde{\pi};\mu) - Q(\tilde{\pi};\hat{\mu})\big) + \big(Q(\tilde{\pi};\hat{\mu}) - Q(\pi^*;\hat{\mu})\big) \\ + \big(Q(\pi^*;\hat{\mu}) - Q(\pi^*;\mu)\big).$$

Since $\tilde{\pi}$ minimizes $Q(\cdot;\hat{\mu})$ over $\Pi_{\alpha,\beta}$, the middle term is non-positive. Therefore

$$Q(\tilde{\pi};\mu) - Q(\pi^*;\mu) \leq 2 \sup_{\pi \in \Pi_{\alpha,\beta}} |Q(\pi;\mu) - Q(\pi;\hat{\mu})|.$$

For any feasible $\pi$, using $\pi_{jk}(x) \geq \alpha$,

$$|Q(\pi; \mu) - Q(\pi; \hat{\mu})| = \left| \mathbb{E} \left[ \sum_{j \neq k} \frac{a_{jk}(X; \mu) - a_{jk}(X; \hat{\mu})}{\pi_{jk}(X)} \right] \right|$$

$$\leq \mathbb{E} \left[ \sum_{j \neq k} \frac{|a_{jk}(X; \mu) - a_{jk}(X; \hat{\mu})|}{\pi_{jk}(X)} \right]$$

$$\leq \frac{1}{\alpha} \mathbb{E} \left[ \sum_{j \neq k} |a_{jk}(X; \mu) - a_{jk}(X; \hat{\mu})| \right].$$

Combining the last two displays yields

$$tr\big(\Sigma(\tilde{\pi})\big) - tr\big(\Sigma(\pi^*)\big) \leq \frac{2}{\alpha} \mathbb{E} \left[ \sum_{j \neq k} |a_{jk}(X; \mu) - a_{jk}(X; \hat{\mu})| \right].$$

By the assumed first-order MSE expansion,

$$R_n(q) = \frac{1}{n} tr\big(\Sigma(q)\big) + o(n^{-1}),$$

for every fixed $q \in \Pi_{\alpha, \beta}$. Therefore

$$R_n(\tilde{\pi}) - R_n(\pi^*) \leq \frac{2}{\alpha n} \mathbb{E} \left[ \sum_{j \neq k} |a_{jk}(X; \mu) - a_{jk}(X; \hat{\mu})| \right] + o(n^{-1}),$$

which proves the first claim.

If, in addition, $a_{jk}(x; \cdot)$ is uniformly $L_a$-Lipschitz and $\|\hat{\mu} - \mu\|_\infty \leq \varepsilon$, then

$$|a_{jk}(x; \hat{\mu}) - a_{jk}(x; \mu)| \leq L_a \varepsilon \qquad \text{for all } x, \ j \neq k,$$

so

$$\mathbb{E} \left[ \sum_{j \neq k} |a_{jk}(X; \hat{\mu}) - a_{jk}(X; \mu)| \right] \leq K(K-1) L_a \varepsilon.$$

Substituting this into the previous bound gives

$$R_n(\tilde{\pi}) - R_n(\pi^*) \leq \frac{2K(K-1)L_a}{\alpha n} \varepsilon + o(n^{-1}),$$

as claimed. $\qquad \square$

## C. Additional examples of GARS

So far, our examples of GARS implicitly focused on settings with binary preference categories (e.g., "$j$ wins" vs. "$k$ wins"). In practice, human feedback often includes richer labels such as ties, or categories indicating that both items are good or both are bad. In this section, we introduce a weighted version of GARS that is compatible with an arbitrary number of categories $C$ and that unifies all examples.

### C.1. Unifying multiple categories via weighted preferences

**Category weights.** Recall that $Y_{jk} \in \{e_1, \ldots, e_C\}$ is a one-hot categorical label for pair $(j, k)$, and that

$$\mu_{jkc}(x) = \mathbb{P}(Y_{jkc} = 1 \mid X = x), \qquad c = 1, \ldots, C$$

denotes the corresponding preference probabilities. For each category $c$, we allow separate weights for the first and the second item in the ordered pair:

$$w^{(1)}, w^{(2)} \in \mathbb{R}^C, \qquad w^{(1)} = (w_1^{(1)}, \ldots, w_C^{(1)}), \quad w^{(2)} = (w_1^{(2)}, \ldots, w_C^{(2)}).$$

Intuitively, $w_c^{(1)}$ measures how much category $c$ contributes in favor of the first item in the ordered pair $(j, k)$, while $w_c^{(2)}$ measures how much category $c$ contributes in favor of the second item.

Given weights $(w^{(1)}, w^{(2)})$, we define *directional weighted scores*

$$s_{jk}^{(1)}(x) = \sum_{c=1}^{C} w_c^{(1)} \mu_{jkc}(x), \qquad\qquad s_{jk}^{(2)}(x) = \sum_{c=1}^{C} w_c^{(2)} \mu_{jkc}(x). \tag{21}$$

Here, $s_{jk}^{(1)}(x)$ aggregates the probabilities of all categories which favor the first item in $(j, k)$, weighted according to $w^{(1)}$, and $s_{jk}^{(2)}(x)$ analogously aggregates categories in favor of the second item.

To account for possible position-bias (or display-order), we symmetrize the directional scores across both orders $(j, k)$ and $(k, j)$ via

$$s_{jk}^{\mathrm{sym}}(x) = \frac{1}{2}\left(s_{jk}^{(1)}(x) + s_{kj}^{(2)}(x)\right), \qquad j \neq k, \tag{22}$$

and we set $s_{jj}^{\mathrm{sym}}(x) = 0$. The quantity $s_{jk}^{\mathrm{sym}}(x)$ can be interpreted as a symmetrized, position-bias-corrected score by how much item $j$ is preferred over item $k$ under context $x$.

**Default weights for common label semantics.** The choice of $(w^{(1)}, w^{(2)})$ is user–specified and encodes how each category is interpreted. In our implementation, we provide the following default weights for common label schemes:

- *Binary preferences* ($C = 2$): categories *win / loss* for the first item.

$$w^{(1)} = (1, 0), \qquad w^{(2)} = (0, 1).$$

- *Ternary preferences* ($C = 3$): categories *win / loss / tie*.

$$w^{(1)} = (1, 0, 1/2), \qquad w^{(2)} = (0, 1, 1/2).$$

- *Quaternary preferences* ($C = 4$): categories *win / loss / both good / both bad*.

$$w^{(1)} = (1, 0, 1, 0), \qquad w^{(2)} = (0, 1, 1, 0).$$

- *Quinary preferences* ($C = 5$): categories *win / loss / both good / both bad / tie*.

$$w^{(1)} = (1, 0, 1, 0, 1/2), \qquad w^{(2)} = (0, 1, 1, 0, 1/2).$$

### C.2. GARS examples for weighted category preferences

The weighted scores $s^{\text{sym}}(x) = \left(s_{jk}^{\text{sym}}(x)\right)_{j,k=1}^{K}$ provide a generic building block from which we can construct different ranking models, all within our GARS framework.

(E1) **Weighted Borda scores.** For item $j$, the weighted Borda score is defined as the average symmetrized score of $j$ against all other items:

$$F_j\big(\mu(x)\big) = \frac{1}{K-1} \sum_{k \neq j} s_{jk}^{\text{sym}}(x), \qquad j = 1, \ldots, K. \tag{23}$$

The corresponding GARS are $\theta = \mathbb{E}[F(\mu(X))] \in \mathbb{R}^K$ with $F(\mu(x)) = (F_1(\mu(x)), \ldots, F_K(\mu(x)))$. When the categories are binary win/loss and we use the default weights above, Eq. (23) coincides with the standard Borda scores.

(E2) **Weighted Bradley–Terry scores.** To obtain a generalized BT–type model, we first form directional "win probabilities" from the weighted scores:

$$p_{jk}^{(1)}(x) = s_{jk}^{(1)}(x), \qquad\qquad\qquad p_{jk}^{(2)}(x) = s_{kj}^{(2)}(x), \tag{24}$$

and define edge logits by averaging over the two display orders,

$$\ell_{jk}(\mu(x)) = \frac{1}{2}\Big(\text{logit}\big(p_{jk}^{(1)}(x)\big) + \text{logit}\big(p_{jk}^{(2)}(x)\big)\Big), \qquad j < k. \tag{25}$$

Let $B \in \mathbb{R}^{\binom{K}{2} \times K}$ be the incidence matrix of the complete graph (row $(j,k)$ equals $e_j - e_k$ for $j < k$), $L_0 = B^\top B$ the unweighted Laplacian, and

$$H = \begin{bmatrix} I_{K-1} \\ -\mathbf{1}_{K-1}^\top \end{bmatrix} \in \mathbb{R}^{K \times (K-1)},$$

so that $\{\phi \in \mathbb{R}^K : \mathbf{1}^\top \phi = 0\} = \{H\alpha : \alpha \in \mathbb{R}^{K-1}\}$ as before. Setting

$$\ell(\mu(x)) = \big(\ell_{jk}(\mu(x))\big)_{j<k} \in \mathbb{R}^{\binom{K}{2}}, \tag{26}$$

we define the generalized BT scores via

$$F(\mu(x)) = H\big(H^\top L_0 H\big)^{-1} H^\top B^\top \ell(\mu(x)) \in \mathbb{R}^K. \tag{27}$$

The corresponding GARS are $\theta = \mathbb{E}[F(\mu(X))]$. In the special case $C = 2$ with binary win/loss labels and default weights, $F(\mu(x))$ reduces to the BT scores from Section 4.1.

(E3) **Weighted rank centrality / PageRank scores.** We can likewise define a weighted version of rank centrality by using the symmetrized scores to build a Markov transition matrix. For each context $x$, let

$$R_{ij}(\mu(x)) = s_{ji}^{\text{sym}}(x), \qquad i \neq j, \tag{28}$$

and $R_{ii}(\mu(x)) = 0$. Intuitively, $R_{ij}$ measures how much item $j$ is preferred over item $i$. We form row–stochastic transition probabilities

$$T_{ij}(\mu(x)) = \frac{R_{ij}(\mu(x))}{\sum_{\ell \neq i} R_{i\ell}(\mu(x))}, \qquad i \neq j, \tag{29}$$

with $T_{ii}(\mu(x)) = 0$. (If the denominator is zero, we may, e.g., define the $i$-th row to be uniform over all $j$.) As in Section 4, we define

$$F(\mu(x)) = \big(I - T(\mu(x))^\top + \mathbf{1}\mathbf{1}^\top\big)^{-1}\mathbf{1} \in \mathbb{R}^K, \tag{30}$$

which yields the stationary distribution of the Markov chain with transition matrix $T(\mu(x))$. The associated GARS are again $\theta = \mathbb{E}[F(\mu(X))]$. When $C = 2$ and the weights correspond to binary win/loss, Eq. ((30)) reduces to the standard rank centrality construction.

**Custom ranking estimands.** The GARS form makes it straightforward to construct new application-specific ranking criteria.

For example, one can define a *thresholded dominance score* that rewards only sufficiently confident wins:

$$F_j^{\text{thr}}(\mu(x)) = \frac{1}{K-1} \sum_{k \neq j} \phi_\tau(\mu_{jk}(x)), \tag{31}$$

where $\mu_{jk}(x)$ is the probability that item $j$ beats item $k$ in context $x$, and $\phi_\tau$ is a thresholding function, e.g.

$$\phi_\tau(u) = \mathbf{1}\{u \geq \tau\} \qquad \text{or} \qquad \phi_\tau(u) = \sigma(\alpha(u - \tau)), \tag{32}$$

for a threshold $\tau > 1/2$, logistic function $\sigma$, and sharpness parameter $\alpha > 0$. The corresponding GARS estimand is

$$\theta_j^{\text{thr}} = \mathbb{E}\big[F_j^{\text{thr}}(\mu(X))\big]. \tag{33}$$

This score emphasizes *clear* pairwise wins rather than marginal ones.

A second example is a *robust / worst-case score*, which favors items that do not have a clear weakness against any competitor:

$$F_j^{\text{rob}}(\mu(x)) = \min_{k \neq j} \mu_{jk}(x), \tag{34}$$

with corresponding estimand

$$\theta_j^{\text{rob}} = \mathbb{E}\big[F_j^{\text{rob}}(\mu(X))\big]. \tag{35}$$

A smooth alternative is the soft minimum

$$F_j^{\text{softrob}}(\mu(x)) = -\frac{1}{\lambda} \log\left( \sum_{k \neq j} \exp(-\lambda\, \mu_{jk}(x)) \right), \qquad \lambda > 0. \tag{36}$$

This score emphasizes robustness by penalizing poor matchups more strongly than average-based criteria.

## D. Derivation of closed-form BT-projection scores.

Here we provide a short derivation of the BT projection scores from Eq. (7). Fix a context $x$, and let

$$r(x) = (r_1(x), \ldots, r_K(x))^\top$$

denote the Bradley–Terry (BT) score vector. For each unordered pair $(j, k)$ with $j < k$, let $B$ be the signed incidence matrix whose $(j, k)$-row satisfies

$$B_{(j,k),i} = \begin{cases} 1, & i = j, \\ -1, & i = k, \\ 0, & \text{otherwise.} \end{cases}$$

Thus,

$$(Br(x))_{jk} = r_j(x) - r_k(x), \qquad j < k.$$

Under the BT model with an order-dependent bias term $b(x)$, the pairwise preference probabilities satisfy

$$\mu_{jk1}(x) = \sigma(r_j(x) - r_k(x) + b(x)), \tag{37}$$
$$\mu_{kj2}(x) = \sigma(r_j(x) - r_k(x) - b(x)), \tag{38}$$

where $\sigma(t) = (1 + \exp(-t))^{-1}$. The second probability uses the reversed ordering. Since $\text{logit}(\sigma(t)) = t$, the symmetrized log-odds are

$$\ell(\mu(x)) = \left( \frac{\text{logit}(\mu_{jk1}(x)) + \text{logit}(\mu_{kj2}(x))}{2} \right)_{j<k} \tag{39}$$

$$= \left( \frac{r_j(x) - r_k(x) + b(x) + r_j(x) - r_k(x) - b(x)}{2} \right)_{j<k} \tag{40}$$

$$= (r_j(x) - r_k(x))_{j<k} \tag{41}$$

$$= Br(x). \tag{42}$$

Thus, under the BT model, the objective in Eq. (3) becomes

$$\min_{\phi: \mathbf{1}^\top \phi = 0} \|B\phi - \ell(\mu(x))\|_2^2 = \min_{\phi: \mathbf{1}^\top \phi = 0} \|B\phi - Br(x)\|_2^2. \tag{43}$$

Since $r(x)$ is feasible and achieves objective value zero, it is a minimizer. With the zero-sum normalization $\mathbf{1}^\top r(x) = 0$, this minimizer is unique. Therefore, Eq. (7) exactly recovers the BT score vector whenever the BT model holds.

**Relation to existing ranking formulations.** The purpose of Eq. (7) is to obtain a closed-form BT-type projection written directly in terms of the preference probabilities $\mu$. Related formulations appear, for example, in HodgeRank-style methods, where an observed pairwise comparison flow is projected onto the score-difference subspace (Jiang et al., 2010). In contrast, Eq. Eq. (7) projects the symmetrized log-odds $\ell(\mu(x))$ and does so in the contextual, nonparametric setting considered here.

# E. Jacobians of weighted GARS examples

Recall that for any GARS of the form

$$\theta(\mathbb{P}) \;=\; \mathbb{E}\big[F(\mu(X))\big] \in \mathbb{R}^d,$$

the efficient influence function and debiased estimator depend on the Jacobian

$$J_{jk}(\mu) \;=\; \frac{\partial F(\mu)}{\partial \mu_{jk}} \;\in\; \mathbb{R}^{d \times C}, \qquad j \neq k,$$

where $\mu_{jk} = (\mu_{jk1}, \ldots, \mu_{jkC})^\top$ is the vector of category probabilities for pair $(j,k)$. Below we derive $J_{jk}(\mu)$ for the three weighted GARS functionals from Section C.1. Throughout, we write $e_r$ for the $r$-th standard basis vector in $\mathbb{R}^K$.

**Preliminaries.** We recall the weighted and symmetrized scores from Eq. ((22)). Given category weights $w^{(1)}, w^{(2)} \in \mathbb{R}^C$, we define

$$s_{jk}^{(1)}(\mu) \;=\; \sum_{c=1}^{C} w_c^{(1)} \mu_{jkc} \;=\; \langle w^{(1)}, \mu_{jk} \rangle,$$

$$s_{jk}^{(2)}(\mu) \;=\; \sum_{c=1}^{C} w_c^{(2)} \mu_{jkc} \;=\; \langle w^{(2)}, \mu_{jk} \rangle,$$

and the symmetrized score

$$s_{jk}^{\mathrm{sym}}(\mu) \;=\; \frac{1}{2}\Big(s_{jk}^{(1)}(\mu) + s_{kj}^{(2)}(\mu)\Big), \qquad j \neq k,$$

with $s_{jj}^{\mathrm{sym}}(\mu) = 0$. Simple differentiation gives, for any $j \neq k$ and $c = 1, \ldots, C$,

$$\frac{\partial s_{jk}^{\mathrm{sym}}(\mu)}{\partial \mu_{jkc}} \;=\; \frac{1}{2}\, w_c^{(1)}, \qquad\qquad\qquad \frac{\partial s_{jk}^{\mathrm{sym}}(\mu)}{\partial \mu_{kjc}} \;=\; \frac{1}{2}\, w_c^{(2)}, \tag{44}$$

and $\partial s_{ab}^{\mathrm{sym}}/\partial \mu_{jkc} = 0$ for all other $(a, b)$.

## E.1. Weighted Borda scores

The weighted Borda scores from Eq. (23) are

$$F_j(\mu) = \frac{1}{K-1} \sum_{k \neq j} s_{jk}^{\mathrm{sym}}(\mu), \qquad j = 1, \ldots, K.$$

Fix a pair $(j, k)$ with $j \neq k$ and a category index $c$. Using Eq. (44) and the definition of $F$, we obtain

$$\frac{\partial F_j(\mu)}{\partial \mu_{jkc}} = \frac{1}{K-1} \frac{\partial s_{jk}^{\mathrm{sym}}(\mu)}{\partial \mu_{jkc}} = \frac{1}{K-1} \cdot \frac{1}{2} w_c^{(1)} = \frac{w_c^{(1)}}{2(K-1)},$$

$$\frac{\partial F_k(\mu)}{\partial \mu_{jkc}} = \frac{1}{K-1} \frac{\partial s_{kj}^{\mathrm{sym}}(\mu)}{\partial \mu_{jkc}} = \frac{1}{K-1} \cdot \frac{1}{2} w_c^{(2)} = \frac{w_c^{(2)}}{2(K-1)},$$

and $\partial F_r(\mu)/\partial \mu_{jkc} = 0$ for all $r \notin \{j, k\}$.

Equivalently, the Jacobian $J_{jk}(\mu) \in \mathbb{R}^{K \times C}$ has only two non-zero rows and can be written compactly as

$$J_{jk}(\mu) = \frac{1}{2(K-1)}\Big(e_j\, w^{(1)\top} + e_k\, w^{(2)\top}\Big), \qquad j \neq k, \tag{45}$$

where $w^{(1)\top}, w^{(2)\top} \in \mathbb{R}^{1 \times C}$ are viewed as row vectors.

### E.2. Weighted Bradley–Terry scores

The weighted BT scores from Eq. (27) are constructed as follows. For each unordered edge $\{a, b\}$ with $a < b$, define

$$\ell_{ab}(\mu) = \frac{1}{2}\Big(\text{logit}\big(s_{ab}^{(1)}(\mu)\big) + \text{logit}\big(s_{ba}^{(2)}(\mu)\big)\Big), \tag{46}$$

and collect these into

$$\ell(\mu) = \big(\ell_{ab}(\mu)\big)_{a < b} \in \mathbb{R}^{\binom{K}{2}}.$$

Let $B \in \mathbb{R}^{\binom{K}{2} \times K}$ be the incidence matrix of the complete graph, $L_0 = B^\top B$ the Laplacian, $H$ the zero-sum basis as in Eq. (27), and define the constant matrix

$$P = H\big(H^\top L_0 H\big)^{-1} H^\top B^\top \in \mathbb{R}^{K \times \binom{K}{2}}.$$

Then

$$F(\mu) = P\,\ell(\mu) \in \mathbb{R}^K$$

gives the (generalized) BT scores.

Fix an ordered pair $(j, k)$ with $j \neq k$ and a category index $c$. The vector $\ell(\mu)$ depends on $\mu_{jk}$ through exactly one edge, namely the unordered edge $\{j, k\}$. Let $e(j, k) \in \{1, \dots, \binom{K}{2}\}$ denote the index of edge $\{j, k\}$ with the convention $a < b$ inside the indexing. Then

$$\frac{\partial F(\mu)}{\partial \mu_{jkc}} = P_{\cdot, e(j,k)}\,\frac{\partial \ell_{j \wedge k,\, j \vee k}(\mu)}{\partial \mu_{jkc}},$$

where $P_{\cdot, e(j,k)} \in \mathbb{R}^K$ is the corresponding column of $P$ and $j \wedge k = \min\{j, k\}$, $j \vee k = \max\{j, k\}$.

We write, for brevity,

$$p_{ab}^{(1)}(\mu) = s_{ab}^{(1)}(\mu) = \langle w^{(1)}, \mu_{ab} \rangle,$$
$$p_{ab}^{(2)}(\mu) = s_{ab}^{(2)}(\mu) = \langle w^{(2)}, \mu_{ab} \rangle,$$

so that $\text{logit}'(p) = 1/\{p(1 - p)\}$ wherever $p \in (0, 1)$. Differentiating Eq. (46) gives

$$\frac{\partial \ell_{ab}(\mu)}{\partial \mu_{abc}} = \frac{1}{2}\,\text{logit}'\big(p_{ab}^{(1)}(\mu)\big)\,\frac{\partial p_{ab}^{(1)}(\mu)}{\partial \mu_{abc}} = \frac{1}{2}\,\frac{w_c^{(1)}}{p_{ab}^{(1)}(\mu)\big(1 - p_{ab}^{(1)}(\mu)\big)},$$

$$\frac{\partial \ell_{ab}(\mu)}{\partial \mu_{bac}} = \frac{1}{2}\,\text{logit}'\big(p_{ba}^{(2)}(\mu)\big)\,\frac{\partial p_{ba}^{(2)}(\mu)}{\partial \mu_{bac}} = \frac{1}{2}\,\frac{w_c^{(2)}}{p_{ba}^{(2)}(\mu)\big(1 - p_{ba}^{(2)}(\mu)\big)},$$

and $\partial \ell_{ab}/\partial \mu_{jkc} = 0$ for all other $(j, k)$.

Thus the Jacobian $J_{jk}(\mu) \in \mathbb{R}^{K \times C}$ is an outer product of the relevant column of $P$ and a category-weighted derivative factor:

- If $j < k$, then $\mu_{jk}$ appears as "first item" in edge $(j, k)$ and

$$J_{jk}(\mu) = P_{\cdot, e(j,k)}\left(\frac{1}{2}\,\frac{w^{(1)\top}}{p_{jk}^{(1)}(\mu)\big(1 - p_{jk}^{(1)}(\mu)\big)}\right), \tag{47}$$

  where the right-hand side is a $K \times C$ outer product.

- If $j > k$, then $\mu_{jk}$ appears as "second item" in edge $(k, j)$ and

$$J_{jk}(\mu) = P_{\cdot, e(k,j)}\left(\frac{1}{2}\,\frac{w^{(2)\top}}{p_{kj}^{(2)}(\mu)\big(1 - p_{kj}^{(2)}(\mu)\big)}\right). \tag{48}$$

In both cases, $J_{jk}(\mu)$ is rank–one as a matrix in $\mathbb{R}^{K \times C}$.

### E.3. Weighted Rank Centrality / PageRank scores

For the weighted rank centrality functional in Eq. (30), we first construct

$$R_{ij}(\mu) = s_{ji}^{\text{sym}}(\mu), \qquad i \neq j, \quad R_{ii}(\mu) = 0,$$

and define row sums

$$d_i(\mu) = \sum_{\ell \neq i} R_{i\ell}(\mu).$$

On the set where $d_i(\mu) > 0$ for all $i$, we define the row-stochastic transition matrix

$$T_{ij}(\mu) = \frac{R_{ij}(\mu)}{d_i(\mu)}, \qquad i \neq j, \quad T_{ii}(\mu) = 0,$$

and set

$$A(\mu) = I - T(\mu)^\top + \mathbf{1}\mathbf{1}^\top.$$

Then the rank centrality scores are

$$F(\mu) = A(\mu)^{-1}\mathbf{1} \in \mathbb{R}^K.$$

Fix an ordered pair $(j, k)$ with $j \neq k$ and category $c$. We first differentiate $R$ and $d$ using Eq. (44). Since $R_{ij}(\mu) = s_{ji}^{\text{sym}}(\mu)$, we obtain

$$\frac{\partial R_{jk}(\mu)}{\partial \mu_{jkc}} = \frac{\partial s_{kj}^{\text{sym}}(\mu)}{\partial \mu_{jkc}} = \frac{1}{2} w_c^{(2)},$$

$$\frac{\partial R_{kj}(\mu)}{\partial \mu_{jkc}} = \frac{\partial s_{jk}^{\text{sym}}(\mu)}{\partial \mu_{jkc}} = \frac{1}{2} w_c^{(1)},$$

and all other partial derivatives $\partial R_{i\ell}/\partial \mu_{jkc}$ vanish. Consequently, the row sums satisfy

$$\frac{\partial d_j(\mu)}{\partial \mu_{jkc}} = \frac{\partial R_{jk}(\mu)}{\partial \mu_{jkc}} = \frac{1}{2} w_c^{(2)},$$

$$\frac{\partial d_k(\mu)}{\partial \mu_{jkc}} = \frac{\partial R_{kj}(\mu)}{\partial \mu_{jkc}} = \frac{1}{2} w_c^{(1)},$$

and $\partial d_i(\mu)/\partial \mu_{jkc} = 0$ for all $i \notin \{j, k\}$.

For $d_i(\mu) > 0$, the quotient rule gives, for any $i, \ell$,

$$\frac{\partial T_{i\ell}(\mu)}{\partial \mu_{jkc}} = \frac{\frac{\partial R_{i\ell}(\mu)}{\partial \mu_{jkc}} d_i(\mu) - R_{i\ell}(\mu) \frac{\partial d_i(\mu)}{\partial \mu_{jkc}}}{d_i(\mu)^2}.$$

Using the structure above, the non-zero derivatives are confined to rows $i = j$ and $i = k$:

$$\frac{\partial T_{jk}(\mu)}{\partial \mu_{jkc}} = \frac{\frac{1}{2} w_c^{(2)} d_j(\mu) - R_{jk}(\mu) \frac{1}{2} w_c^{(2)}}{d_j(\mu)^2} = \frac{1}{2} w_c^{(2)} \frac{d_j(\mu) - R_{jk}(\mu)}{d_j(\mu)^2},$$

$$\frac{\partial T_{j\ell}(\mu)}{\partial \mu_{jkc}} = -\frac{R_{j\ell}(\mu) \frac{1}{2} w_c^{(2)}}{d_j(\mu)^2}, \qquad \ell \neq k,$$

$$\frac{\partial T_{kj}(\mu)}{\partial \mu_{jkc}} = \frac{\frac{1}{2} w_c^{(1)} d_k(\mu) - R_{kj}(\mu) \frac{1}{2} w_c^{(1)}}{d_k(\mu)^2} = \frac{1}{2} w_c^{(1)} \frac{d_k(\mu) - R_{kj}(\mu)}{d_k(\mu)^2},$$

$$\frac{\partial T_{k\ell}(\mu)}{\partial \mu_{jkc}} = -\frac{R_{k\ell}(\mu) \frac{1}{2} w_c^{(1)}}{d_k(\mu)^2}, \qquad \ell \neq j,$$

and $\partial T_{i\ell}(\mu)/\partial \mu_{jkc} = 0$ whenever $i \notin \{j, k\}$.

Since $A(\mu) = I - T(\mu)^\top + \mathbf{1}\mathbf{1}^\top$, only the transpose $T(\mu)^\top$ depends on $\mu$, and therefore

$$\frac{\partial A(\mu)}{\partial \mu_{jkc}} = -\frac{\partial T(\mu)^\top}{\partial \mu_{jkc}},$$

i.e., the $(a, b)$-entry of $\partial A(\mu)/\partial \mu_{jkc}$ equals $-\partial T_{ba}(\mu)/\partial \mu_{jkc}$.

Finally, differentiating $F(\mu) = A(\mu)^{-1}\mathbf{1}$ with respect to $\mu_{jkc}$ using the identity $\partial A^{-1} = -A^{-1}(\partial A)A^{-1}$ yields

$$\frac{\partial F(\mu)}{\partial \mu_{jkc}} = -A(\mu)^{-1}\left(\frac{\partial A(\mu)}{\partial \mu_{jkc}}\right) F(\mu) \in \mathbb{R}^K. \tag{49}$$

Thus the Jacobian $J_{jk}(\mu) \in \mathbb{R}^{K \times C}$ has columns

$$J_{jk}(\mu)_{\cdot c} = \frac{\partial F(\mu)}{\partial \mu_{jkc}} = -A(\mu)^{-1}\left(\frac{\partial A(\mu)}{\partial \mu_{jkc}}\right) F(\mu), \qquad c = 1, \ldots, C,$$

with $\partial A(\mu)/\partial \mu_{jkc}$ determined via the explicit expressions for $\partial T_{i\ell}(\mu)/\partial \mu_{jkc}$ above. In practice, these formulas can be implemented by (1) computing $A(\mu)$ and $F(\mu)$, (2) constructing the sparse matrix $\partial A(\mu)/\partial \mu_{jkc}$, and (3) solving the linear system in Eq. (49) for each $c$.

# F. Efficient inference under the Bradley–Terry restricted model

This appendix derives the semiparametrically efficient influence function (EIF), the corresponding one–step estimator, and the A–optimal acquisition rule when the data–generating process is known to satisfy the Bradley–Terry (BT) model. Throughout this section, we focus on binary categories $\mathcal{C} = \{1, 2\}$.

## F.1. BT projection as a misspecification–robust target map

Recall the construction in Sec. 4.1. Let $B \in \mathbb{R}^{\binom{K}{2} \times K}$ be the (unordered) incidence matrix whose row $(j, k)$ equals $e_j - e_k$ for $j < k$, and let $L_0 = B^\top B$ be the (unweighted) graph Laplacian. Define the symmetrized log–odds vector

$$\ell(\mu(x)) = \left( \frac{\text{logit}(\mu_{jk1}(x)) + \text{logit}(\mu_{kj2}(x))}{2} \right)_{j<k} \in \mathbb{R}^{\binom{K}{2}}, \tag{50}$$

and let

$$H = \begin{bmatrix} I_{K-1} \\ -\mathbf{1}_{K-1}^\top \end{bmatrix}, \qquad \{r \in \mathbb{R}^K : \mathbf{1}^\top r = 0\} = \{H\alpha : \alpha \in \mathbb{R}^{K-1}\}. \tag{51}$$

The main text defines the *BT projection map*

$$F_{\text{BT}}(\mu(x)) = H (H^\top L_0 H)^{-1} H^\top B^\top \ell(\mu(x)) \in \mathbb{R}^K. \tag{52}$$

Importantly, $F_{\text{BT}}$ is well-defined for any $\mu(x) \in (0, 1)^{K \times K \times 2}$ (after truncation to avoid $\text{logit}(0), \text{logit}(1)$) and does *not* assume that the BT model holds. If BT is misspecified, the map $F_{\text{BT}}(\mu(x))$ returns the $L^2$-projection of the edge log–odds vector $\ell(\mu(x))$ onto the BT subspace $\{Br : \mathbf{1}^\top r = 0\}$, yielding a principled "BT-like" score vector even under misspecification.

## F.2. Assuming the BT model holds

Now assume the BT model from Eq. (4) holds:

$$\mu_{jk1}(x) = \sigma(r_j(x) - r_k(x) + b(x)), \qquad \sum_{m=1}^K r_m(x) = 0,$$

where $b(x)$ is a (possibly unknown) position-bias term. Under this model,

$$\text{logit}(\mu_{jk1}(x)) = r_j(x) - r_k(x) + b(x), \qquad \text{logit}(\mu_{kj2}(x)) = r_j(x) - r_k(x) - b(x),$$

so the symmetrization in Eq. (50) cancels $b(x)$ and yields

$$\ell_{jk}(\mu(x)) = r_j(x) - r_k(x) \qquad (j < k). \tag{53}$$

Consequently, the BT projection is exact:

$$F_{\text{BT}}(\mu(x)) = r(x), \qquad \text{and hence} \qquad \theta = \mathbb{E}[r(X)] = \mathbb{E}[F_{\text{BT}}(\mu(X))]. \tag{54}$$

## F.3. EIF under the BT restriction, parametrized by $(\mu, \pi)$

The EIF in Theorem 5.1 applies to the *unrestricted* (nonparametric) model for $\mu$ and uses the Jacobian $J_{jk}(\mu) = \nabla_{\mu_{jk}} F(\mu)$. Under the BT restriction, $\mu(x)$ is constrained to a lower-dimensional manifold. The efficient EIF in this *restricted* model retains the same AIPW/EIF-like structure as Eq. (13), but with the *same target map* $F_{\text{BT}}$ and a *different* "Jacobian" that accounts for the restriction.

To define it, first map $\mu(x)$ to the BT-consistent (bias-cancelled) pairwise probabilities

$$\bar{\mu}_{jk}(x) = \sigma(\ell_{jk}(\mu(x))), \qquad (j < k), \tag{55}$$

which equals $\sigma(r_j(x) - r_k(x))$ when BT holds by (53). Let $W(\mu(x)) \in \mathbb{R}^{\binom{K}{2} \times \binom{K}{2}}$ be diagonal with entries

$$W_{(j,k),(j,k)}(\mu(x)) = \bar{\mu}_{jk}(x)(1 - \bar{\mu}_{jk}(x)), \qquad (j < k), \tag{56}$$

and define the *BT information Laplacian*

$$L_{\mathrm{BT}}(\mu(x)) = B^\top W(\mu(x))B \in \mathbb{R}^{K \times K}. \tag{57}$$

This Laplacian is singular in the all-ones direction; on the constraint subspace $\{r : \mathbf{1}^\top r = 0\}$ its inverse is

$$L_{\mathrm{BT}}^\dagger(\mu(x)) = H\big(H^\top L_{\mathrm{BT}}(\mu(x))H\big)^{-1}H^\top. \tag{58}$$

For an ordered pair $(j, k)$, let $b_{jk} = e_j - e_k$. In the binary case, define the BT-restricted Jacobian blocks $J_{jk}^{\mathrm{BT}}(\mu(x)) \in \mathbb{R}^{K \times 2}$ by

$$J_{jk}^{\mathrm{BT}}(\mu(x)) = \left[\; \frac{1}{2}\,L_{\mathrm{BT}}^\dagger(\mu(x))\,b_{jk}\;,\; -\frac{1}{2}\,L_{\mathrm{BT}}^\dagger(\mu(x))\,b_{jk}\;\right]. \tag{59}$$

With this choice, the BT-restricted EIF can be written in the same template as Eq. (13):

**Theorem F.1** (EIF under the BT restriction). *Assume $\mathcal{C} = \{1, 2\}$ and the BT model (4) holds. Then, the efficient influence function for $\theta = \mathbb{E}[r(X)] = \mathbb{E}[F_{\mathrm{BT}}(\mu(X))] \in \mathbb{R}^K$ in the BT-restricted model is*

$$\phi_{\mathrm{BT}}(O, \eta, \theta) = F_{\mathrm{BT}}\big(\mu(X)\big) - \theta + \sum_{j \neq k} \frac{S_{jk}}{\pi_{jk}(X)}\, J_{jk}^{\mathrm{BT}}\big(\mu(X)\big)\,\big(Y_{jk} - \mu_{jk}(X)\big). \tag{60}$$

*Equivalently, using the scalar residual $\varepsilon_{jk}(X) = Y_{jk1} - \mu_{jk1}(X)$, one may write*

$$\phi_{\mathrm{BT}}(O, \eta, \theta) = r(X) - \theta + \sum_{j \neq k} \frac{S_{jk}}{\pi_{jk}(X)}\, L_{\mathrm{BT}}^\dagger\big(\mu(X)\big)\,b_{jk}\,\varepsilon_{jk}(X).$$

**Same $F$, different $J$.** Comparing Eq. (60) with Eq. (13), the BT restriction does *not* change the overall EIF structure; it changes (i) the choice of target map $F$ (here: $F_{\mathrm{BT}}$) and (ii) the "Jacobian" blocks multiplying the residual. In the unrestricted model, these blocks are the literal Jacobian $J_{jk}(\mu) = \nabla_{\mu_{jk}} F(\mu)$; in the BT-restricted model, $J_{jk}^{\mathrm{BT}}$ in Eq. (59) is the Riesz representer (the projection of the unrestricted gradient onto the BT tangent space).

### F.4. Efficiency bounds: unrestricted vs. BT-restricted

Let $\Sigma_{\mathrm{NP}}(\pi)$ denote the semiparametric efficiency bound from Theorem 5.1 when using the BT projection target $F = F_{\mathrm{BT}}$ but treating $\mu$ as unrestricted (so $J_{jk} = \nabla_{\mu_{jk}} F_{\mathrm{BT}}$). Let $\Sigma_{\mathrm{BT}}(\pi)$ denote the efficiency bound under the BT restriction:

$$\Sigma_{\mathrm{BT}}(\pi) = \mathbb{E}\big[\phi_{\mathrm{BT}}(O, \eta, \theta)\,\phi_{\mathrm{BT}}(O, \eta, \theta)^\top\big].$$

Since the BT-restricted model has a smaller tangent space, its EIF is the orthogonal projection of the unrestricted EIF onto that tangent space, and therefore the bound can only improve:

$$\Sigma_{\mathrm{BT}}(\pi) \preceq \Sigma_{\mathrm{NP}}(\pi) \qquad \text{(positive semidefinite order).} \tag{61}$$

Equivalently, for any contrast $a \in \mathbb{R}^K$, $\mathrm{Var}\big(a^\top \phi_{\mathrm{BT}}\big) \leq \mathrm{Var}\big(a^\top \phi_{\mathrm{NP}}\big)$.

### F.5. BT-restricted debiased estimator

Using cross-fitted nuisance estimates $\hat{\eta} = (\hat{\mu}, \hat{\pi})$ as in Sec. 5, the one-step estimator is obtained by plugging $\hat{\mu}, \hat{\pi}$ into Eq. (60) and averaging:

$$\hat{\theta}_{\mathrm{EIF, BT}} = \frac{1}{n}\sum_{i=1}^n \left[ F_{\mathrm{BT}}\big(\hat{\mu}(x_i)\big) + \sum_{j \neq k} \frac{s_{i,jk}}{\hat{\pi}_{jk}(x_i)}\, J_{jk}^{\mathrm{BT}}\big(\hat{\mu}(x_i)\big)\,\big(y_{i,jk} - \hat{\mu}_{jk}(x_i)\big) \right]. \tag{62}$$

For implementation, one may compute $L_{\mathrm{BT}}^\dagger(\hat{\mu}(x_i))$ via Eq. (57)–Eq. (58) and then form $J_{jk}^{\mathrm{BT}}(\hat{\mu}(x_i))$ using Eq. (59). A covariance estimator is the empirical covariance of the cross-fitted influence values, analogously to Theorem 5.1.

## F.6. A-optimal acquisition under the BT restriction

Define the BT-restricted efficiency bound $\Sigma_{\mathrm{BT}}(\pi)$ as above, and consider the same feasible class $\Pi_{\alpha,\beta}$ from Sec. 6. Under Assumption B.1, minimizing $\mathrm{tr}(\Sigma_{\mathrm{BT}}(\pi))$ yields the same closed-form structure as Theorem 6.2, but with $J_{jk}^{\mathrm{BT}}$ in place of $J_{jk}$:

**Corollary F.2** (A-optimal policy under BT restriction). *Assume Assumption B.1 and $\mathcal{C} = \{1, 2\}$. Any A-optimal policy for the BT-restricted bound satisfies, for a.e. $x$ and all $j \neq k$,*

$$\pi_{jk}^*(x) = \mathrm{clip}_{[\alpha,1]} \sqrt{\frac{\mathrm{tr}\Big( J_{jk}^{\mathrm{BT}}(\mu(x))\, V_{jk}(\mu(x))\, \big( J_{jk}^{\mathrm{BT}}(\mu(x)) \big)^{\top} \Big)}{\lambda_A\, c_{jk}}}, \tag{63}$$

*where $V_{jk}(\mu(x)) = \mathrm{Diag}(\mu_{jk}(x)) - \mu_{jk}(x)\mu_{jk}(x)^{\top}$ and $\lambda_A \geq 0$ is chosen so that $\mathbb{E}[\sum_{j\neq k} c_{jk}\pi_{jk}^*(X)] = \beta$.*

In the binary case, Eq. (63) simplifies to

$$\mathrm{tr}\Big( J_{jk}^{\mathrm{BT}}(\mu(x))\, V_{jk}(\mu(x))\, \big( J_{jk}^{\mathrm{BT}}(\mu(x)) \big)^{\top} \Big) = \mu_{jk1}(x)\big(1 - \mu_{jk1}(x)\big) \big\| L_{\mathrm{BT}}^{\dagger}(\mu(x))\, b_{jk} \big\|_2^2,$$

so the BT-restricted A-optimal rule upweights pairs whose outcomes are both (i) noisy (large Bernoulli variance) and (ii) influential for the BT score vector as measured by the weighted-Laplacian pseudoinverse.

# G. Extension to labeling at most one pair of items per context

In this appendix, we consider the design where, for each prompt $X$, *at most one* unordered pair $(j < k)$ is selected for labeling; that is,

$$\sum_{j<k} S_{jk} \in \{0,1\} \qquad \text{a.s.}$$

We interpret $\sum_{j<k} S_{jk} = 0$ as "no label is collected for this context."

**Efficient inference unchanged.** The target remains $\theta = \mathbb{E}[F(\mu(X))]$. The efficient influence function and the one-step estimator of Theorem 5.1, as well as the covariance estimator and confidence regions in Eq. (17), are *unchanged in form*. The only change is the selection (propensity) model and the feasible set for optimal labeling policies. Under at-most-one selection, the propensity vector $\pi(x) = \{\pi_{jk}(x)\}_{j<k}$ satisfies a *sub-simplex* constraint,

$$\pi_{jk}(x) = \mathbb{P}(S_{jk} = 1 \mid X = x), \qquad \sum_{j<k} \pi_{jk}(x) \le 1.$$

It is convenient to introduce the "no-label" probability

$$\pi_0(x) \;=\; \mathbb{P}\Big(\sum_{j<k} S_{jk} = 0 \;\Big|\; X = x\Big) \;=\; 1 - \sum_{j<k} \pi_{jk}(x).$$

In Stage 1, we can therefore fit the selection model as a $\left(\binom{K}{2} + 1\right)$–class (softmax) regression with classes

$$\{(j,k) : j < k\} \;\cup\; \{\text{none}\},$$

using cross-fitting to obtain out-of-fold predictions $\widehat{\pi}^{(-v)}(x_i)$ (and $\widehat{\pi}_0^{(-v)}(x_i)$ if needed). If the labeling policy is known by design, we set $\widehat{\pi} \equiv \pi$.

## At-most-one selection model for optimal labeling

We replace Assumption B.1 by the following.

**Assumption G.1** (One-hot-or-zero selection)**.** For $x \in \mathcal{X}$ almost everywhere (a.e.), the selection vector $S = \{S_{jk}\}_{j<k}$ satisfies $\sum_{j<k} S_{jk} \in \{0,1\}$ and has conditional distribution

$$\mathbb{P}(S = s \mid X = x) \;=\; \pi_0(x)^{s_0} \prod_{j<k} \pi_{jk}(x)^{s_{jk}}, \qquad s_0 := 1 - \sum_{j<k} s_{jk} \in \{0,1\},$$

where $\pi_{jk}(x) = \mathbb{P}(S_{jk} = 1 \mid X = x)$, $\pi_0(x) = 1 - \sum_{j<k} \pi_{jk}(x)$, and

$$\sum_{j<k} \pi_{jk}(x) \le 1, \qquad \pi_{jk}(x) \in [\alpha, 1]$$

for some user-chosen $\alpha \in [0, 1/\binom{K}{2}]$.

Let $V_{jk}(\mu) = \mathrm{Diag}(\mu_{jk}) - \mu_{jk}\mu_{jk}^\top \in \mathbb{R}^{C \times C}$ and define

$$W_{jk}(x) \;=\; \mathrm{tr}\Big(J_{jk}(\mu(x))\, V_{jk}(\mu(x))\, J_{jk}(\mu(x))^\top\Big) \;\in\; \mathbb{R}_{\ge 0},$$

where $J_{jk}(\mu)$ is the Jacobian block defined below Theorem 5.1. Under Assumption G.1, the design-dependent contribution to the (A-optimal) asymptotic variance remains

$$\mathcal{L}(\pi) \;=\; \mathbb{E}\Big[\sum_{j<k} \frac{W_{jk}(X)}{\pi_{jk}(X)}\Big]. \tag{64}$$

(Only the feasible set for $\pi$ changes relative to the independent-Bernoulli case.)

**Feasible policies.** Let pair costs be $c_{jk} > 0$ and let $\beta > 0$ be an average-cost budget. We consider

$$\Pi_{\alpha,\beta}^{\leq 1} = \Big\{\pi : \sum_{j<k} \pi_{jk}(x) \leq 1 \text{ a.e.}, \ \pi_{jk}(x) \in [\alpha, 1], \ \mathbb{E}\big[\sum_{j<k} c_{jk}\pi_{jk}(X)\big] \leq \beta\Big\}.$$

Feasibility requires $\alpha \binom{K}{2} \leq 1$ and, if $\alpha > 0$, also $\beta \geq \mathbb{E}[\sum_{j<k} c_{jk}\alpha] = \alpha \sum_{j<k} c_{jk}$.

**Definition G.2** (A-optimality under at-most-one selection). A policy $\pi^* \in \Pi_{\alpha,\beta}^{\leq 1}$ is A-optimal if it minimizes $\text{tr}(\Sigma(\pi))$, equivalently it minimizes $\mathcal{L}(\pi)$ in Eq. (64).

**Theorem G.3** (A-optimal labeling with at-most-one selection). *Suppose $\Pi_{\alpha,\beta}^{\leq 1} \neq \emptyset$. Then there exists $\lambda \geq 0$ and a measurable function $\nu : \mathcal{X} \to \mathbb{R}_{\geq 0}$ such that an A-optimal policy $\pi^*$ satisfies, for a.e. $x$ and all $j < k$,*

$$\pi_{jk}^*(x) = \text{clip}_{[\alpha,1]}\left(\sqrt{\frac{W_{jk}(x)}{\lambda c_{jk} + \nu(x)}}\right), \tag{65}$$

*with complementary slackness*

$$\nu(x)\Big(\sum_{j<k} \pi_{jk}^*(x) - 1\Big) = 0, \qquad \sum_{j<k} \pi_{jk}^*(x) \leq 1, \qquad \nu(x) \geq 0, \tag{66}$$

*and with $\lambda$ chosen so that*

$$\mathbb{E}\Big[\sum_{j<k} c_{jk}\,\pi_{jk}^*(X)\Big] = \beta \quad \text{if the budget is active,} \qquad \lambda = 0 \text{ if the budget is slack.}$$

*The induced "no-label" probability is $\pi_0^*(x) = 1 - \sum_{j<k} \pi_{jk}^*(x) \in [0,1]$.*

*If $\alpha = 0$ and the per-context constraint is active (i.e., $\sum_{j<k} \pi_{jk}^*(x) = 1$), then the interior solution reduces to a square-root rule:*

$$\pi_{jk}^*(x) = \frac{\sqrt{W_{jk}(x)}}{\sum_{a<b} \sqrt{W_{ab}(x)}} \qquad (\lambda = 0).$$

*More generally, when $\alpha = 0$ and $\nu(x) = 0$ (i.e., the constraint $\sum_{j<k} \pi_{jk}(x) \leq 1$ is slack at $x$), the interior KKT solution is*

$$\pi_{jk}^*(x) = \sqrt{\frac{W_{jk}(x)}{\lambda c_{jk}}} \quad \text{and} \quad \pi_0^*(x) = 1 - \sum_{j<k} \sqrt{\frac{W_{jk}(x)}{\lambda c_{jk}}}.$$

*Proof sketch.* The objective $\mathcal{L}(\pi)$ is convex in $\pi$ over the feasible set. Consider the Lagrangian with multipliers $\lambda \geq 0$ (budget), $\nu(x) \geq 0$ (per-$x$ constraint $\sum \pi \leq 1$), and $\tau_{jk}(x) \geq 0$ (lower bounds $\pi_{jk} \geq \alpha$). Stationarity on the active set where $\pi_{jk}(x) > \alpha$ gives

$$-\frac{W_{jk}(x)}{\pi_{jk}(x)^2} + \lambda c_{jk} + \nu(x) = 0,$$

yielding Eq. (65); clamping handles the lower bound. Complementary slackness for $\nu(x)$ yields Eq. (66). Finally, $\lambda$ is chosen to satisfy the (possibly active) budget constraint. $\square$

**Algorithm: water-filling on the capped sub-simplex.** Given samples $\{x_i\}_{i=1}^n$, costs $\{c_{jk}\}$, lower bound $\alpha$, and tolerances $\varepsilon_{\text{in}}, \varepsilon_{\text{out}} > 0$:

1. **Outer loop (budget via bisection in $\lambda \geq 0$).** Initialize $0 = \lambda_{\text{lo}} < \lambda_{\text{hi}}$ large enough that the budget is satisfied. While the estimated average cost deviates from $\beta$ by more than $\varepsilon_{\text{out}}$:

    (a) Set $\lambda \leftarrow (\lambda_{\text{lo}} + \lambda_{\text{hi}})/2$.
    (b) **Per-context water-filling.** For each $x_i$:

    i. Define the decreasing function

$$g_i(\nu) = \sum_{j<k} \max\left\{\alpha, \sqrt{\frac{W_{jk}(x_i)}{\lambda c_{jk} + \nu}}\right\}.$$

    ii. If $g_i(0) \leq 1$, set $\nu(x_i) \leftarrow 0$ (the constraint is slack), and define

$$\pi_{jk}(x_i; \lambda) = \max\left\{\alpha, \sqrt{\frac{W_{jk}(x_i)}{\lambda c_{jk}}}\right\}.$$

Otherwise, find $\nu(x_i) > 0$ by bisection so that $g_i(\nu(x_i)) = 1$, and set

$$\pi_{jk}(x_i; \lambda) = \max\left\{\alpha, \sqrt{\frac{W_{jk}(x_i)}{\lambda c_{jk} + \nu(x_i)}}\right\}.$$

    iii. Set $\pi_0(x_i; \lambda) = 1 - \sum_{j<k} \pi_{jk}(x_i; \lambda)$.

  (c) Estimate the average cost

$$\widehat{C}(\lambda) = \frac{1}{n} \sum_{i=1}^{n} \sum_{j<k} c_{jk}\, \pi_{jk}(x_i; \lambda).$$

If $\widehat{C}(\lambda) > \beta$, increase $\lambda_{\text{lo}} \leftarrow \lambda$; else decrease $\lambda_{\text{hi}} \leftarrow \lambda$.

2. **Output.** The policy $\pi^*(\cdot) = \pi(\cdot; \lambda)$ from the last iteration (and $\pi_0^*(x) = 1 - \sum_{j<k} \pi_{jk}^*(x)$).

**Remarks.** (i) Feasibility requires $\alpha\binom{K}{2} \leq 1$; if equality holds then $\sum_{j<k} \pi_{jk}(x) = 1$ a.e. and $\pi_0(x) \equiv 0$. (ii) When $c_{jk} \equiv 1$, the budget $\beta$ can be interpreted as an *average labeling rate* (since $\mathbb{E}[\sum_{j<k} \pi_{jk}(X)] \leq \beta$ and $\sum_{j<k} \pi_{jk}(x) \leq 1$). (iii) All results are stated for unordered pairs ($j < k$); if using ordered pairs, replace $\sum_{j<k}$ by $\sum_{j \neq k}$ and adjust $W_{jk}$ accordingly.

# H. Additional results for optimal data acquisition

**Beyond A-optimality.** In Sec. 6 we focus on *A-optimal* labeling, i.e., minimizing $\mathrm{tr}(\Sigma(\pi))$, which targets small *marginal* variances (and thus short componentwise CIs). Another standard criterion from optimal experimental design is *D-optimality* (Atkinson et al., 2007), which, in our case, would target the *joint* uncertainty of the full GARS vector.

**Definition H.1** (D-optimal labeling policy). A feasible labeling policy $\pi^D$ is *D-optimal* if

$$\pi^D \in \arg\min_{\pi \in \Pi_{\alpha,\beta}} \det\big(\Sigma(\pi)\big) \qquad \text{equivalently} \qquad \pi^D \in \arg\min_{\pi \in \Pi_{\alpha,\beta}} \log\det\big(\Sigma(\pi)\big). \tag{67}$$

**Interpretation.** For the confidence ellipsoid in Eq. (17), the volume is proportional to $\det(\Sigma(\pi))^{1/2}$ (up to constants independent of $\pi$). Hence, D-optimal labeling tends to yield small *joint* uncertainty (small ellipsoid volume), which is often desirable when one cares about the entire GARS vector simultaneously.

**Theorem H.2** (D-optimal labeling policy). *Assume $\Pi_{\alpha,\beta} \neq \emptyset$ and Assumption B.1. Define, for each ordered pair $(j,k)$ and context $x$,*

$$M_{jk}(x) := J_{jk}(\mu(x))\, V_{jk}(\mu(x))\, J_{jk}(\mu(x))^\top \ \in \ \mathbb{R}^{d \times d}, \qquad V_{jk}(\mu(x)) = \mathrm{Var}(Y_{jk} \mid X = x).$$

*Then, any D-optimal policy $\pi^D$ satisfies, for a.e. $x$ and all $j \neq k$,*

$$\pi_{jk}^D(x) = \mathrm{clip}_{[\alpha,1]} \sqrt{\frac{\mathrm{tr}\big(\Sigma(\pi^D)^{-1}\, M_{jk}(x)\big)}{\lambda_D\, c_{jk}}}, \tag{68}$$

*where $\mathrm{clip}_{[\alpha,1]}(t) = \min\{1, \max\{\alpha, t\}\}$ and $\lambda_D \geq 0$ is chosen so that $\mathbb{E}\big[\sum_{j \neq k} c_{jk} \pi_{jk}^D(X)\big] = \beta$ whenever the budget binds. The characterization in Eq. (68) is a fixed point since $\Sigma(\pi^D)$ depends on $\pi^D$.*

**Computation (fixed-point iteration).** Unlike the A-optimal rule in Theorem 6.2, the D-optimal rule in Eq. (68) depends on $\Sigma(\pi^D)^{-1}$ and is therefore not closed-form. A simple solver is a fixed-point iteration:

1. Initialize $\pi^{(0)}$ (e.g., the A-optimal policy from Theorem 6.2).

2. For $t = 0, 1, 2, \ldots$:

   (a) Compute $\Sigma(\pi^{(t)})$ (e.g., via the population formula if available, or via an empirical estimate using samples $\{x_i\}_{i=1}^n$).
   (b) Update

   $$\tilde{\pi}_{jk}^{(t+1)}(x) = \mathrm{clip}_{[\alpha,1]} \sqrt{\frac{\mathrm{tr}\big(\Sigma(\pi^{(t)})^{-1}\, M_{jk}(x)\big)}{\lambda\, c_{jk}}}$$

   with $\lambda$ chosen by one-dimensional bisection so that the budget constraint holds.
   (c) Set $\pi^{(t+1)} \leftarrow \tilde{\pi}^{(t+1)}$ and stop when the iterates stabilize.

# I. Simultaneous confidence intervals

This appendix details the covariance estimator and the simultaneous confidence intervals used in our experiments. Throughout, let $\widehat{\theta} := \widehat{\theta}_{\mathrm{EIF}}$ denote the debiased estimator from Eq. (14), and let $\phi(O, \eta, \theta)$ be the EIF from Eq. (13).

## I.1. Estimating the asymptotic covariance

For each observation $O_i$, define the estimated influence vector

$$\widehat{\phi}_i := \phi\big(O_i, \widehat{\eta}, \widehat{\theta}\big) \in \mathbb{R}^d, \qquad i = 1, \ldots, n,$$

where $\widehat{\eta} = (\widehat{\mu}, \widehat{\pi})$ denotes the (cross-fitted) nuisance estimates. We estimate the asymptotic covariance of $\sqrt{n}\,(\widehat{\theta} - \theta)$ by

$$\widehat{\Sigma} := \frac{1}{n} \sum_{i=1}^{n} \widehat{\phi}_i \, \widehat{\phi}_i^{\top},$$

which matches the estimator used in Sec. 5. (Optionally, one may replace $\widehat{\phi}_i$ by $\widehat{\phi}_i - \bar{\phi}$ with $\bar{\phi} = \frac{1}{n} \sum_i \widehat{\phi}_i$; this does not change the asymptotics but can improve finite-sample stability.) The asymptotic covariance of $\widehat{\theta}$ itself is $\widehat{\Sigma}/n$, and the componentwise standard errors are

$$\widehat{\mathrm{se}}_j := \sqrt{\widehat{\Sigma}_{jj}/n}, \qquad j = 1, \ldots, d.$$

## I.2. Simultaneous coordinatewise confidence intervals

We report simultaneously valid confidence intervals of the generic form

$$\mathrm{CI}_j(\alpha) = \widehat{\theta}_j \pm c_\alpha \, \widehat{\mathrm{se}}_j, \qquad j = 1, \ldots, d,$$

where the critical value $c_\alpha$ is chosen to achieve joint (family-wise) coverage at level $1 - \alpha$ under the multivariate normal approximation from Theorem 5.1. We use the following two choices of $c_\alpha$.

**Gaussian-max (rectangular) intervals.** Let $\widehat{D} := \mathrm{diag}(\widehat{\Sigma})$ and define the estimated correlation matrix

$$\widehat{R} := \widehat{D}^{-1/2}\, \widehat{\Sigma}\, \widehat{D}^{-1/2}.$$

Draw $Z^{(b)} \sim \mathcal{N}(0, \widehat{R})$ for $b = 1, \ldots, B$ and compute

$$T^{(b)} := \max_{1 \le j \le d} \big|Z_j^{(b)}\big|.$$

Set $c_\alpha$ to the empirical $(1 - \alpha)$-quantile of $\{T^{(b)}\}_{b=1}^{B}$. We use $B = 100{,}000$ Monte-Carlo draws.

**Bonferroni intervals.** Set

$$c_\alpha := \Phi^{-1}\Big(1 - \frac{\alpha}{2d}\Big),$$

which yields (asymptotically) family-wise error control under Bonferroni correction.

**(Optional) marginal intervals.** For non-simultaneous intervals one may use $c_\alpha = \Phi^{-1}(1 - \alpha/2)$; we do not report these in experiments.

## I.3. Ellipsoidal confidence region

The same covariance estimate yields the $(1 - \alpha)$ confidence ellipsoid stated in Eq. (17):

$$\mathcal{E} = \Big\{\vartheta \in \mathbb{R}^d : \; n\,(\widehat{\theta} - \vartheta)^{\top}\widehat{\Sigma}^{-1}(\widehat{\theta} - \vartheta) \le \chi_{d,\, 1-\alpha}^2\Big\}.$$

This ellipsoid provides a joint confidence region for $\theta$, while the Gaussian-max and Bonferroni constructions above provide axis-aligned simultaneous intervals for the individual coordinates.

## J. Intuition: connection to causal inference

This section clarifies the connection between our setting and standard causal estimation problems. In causal inference, the average treatment effect is

$$\tau = \mathbb{E}[Y(1) - Y(0)],$$

where $Y(1)$ and $Y(0)$ denote the potential outcomes under treatment and control. The key feature is that the full potential-outcome vector is never observed for any unit: for each unit, only one component is observed. However, the estimand is not the full distribution of the potential outcomes, but the expectation of a linear functional of them.

A closely analogous structure appears in our ranking setting. For example, the simple Borda score for item $j$ can be written as

$$\theta_j = \mathbb{E}\left[\frac{1}{2(K-1)} \sum_{k \neq j} (Y_{jk1} + Y_{kj2})\right],$$

where $Y_{jk1}$ and $Y_{kj2}$ denote the pairwise comparison outcomes under the two relevant orderings. As in the causal setting, we do not observe all components of the outcome vector $Y$. Nevertheless, the target is only the expectation of a linear functional of $Y$, not the full joint distribution of all comparison outcomes.

Under identifiability (i.e., under consistency, positivity, and ignorability (Rubin, 1974)), the ATE can be written as

$$\tau = \mathbb{E}[E[Y|X, A = 1] - E[Y|X, A = 0]]$$

and can thus be estimated from observed data. In our setting, the corresponding formula for the Borda score becomes

$$\theta_j = \mathbb{E}\left[\frac{1}{2(K-1)} \sum_{k \neq j} (\mu_{jk1}(X) + \mu_{kj2}(X))\right] = \mathbb{E}[F(\mu(X))].$$

This perspective makes the analogy to causal inference explicit. The Borda score is marginalized sum of two conditional expectations, while the ATE is a closely related contrast of two expectations. As a consequence, debiased estimators of Borda-score are of similar form as AIPTW for ATE (Robins et al., 1994).

# K. Details on nuisance estimation

This section provides details on training and tuning the nuisance estimators used for the experiments[2].

## K.1. Overview

Across all experiments we estimate the nuisance functions $\mu_{jkc}(x) = \mathbb{P}(Y_{jk} = c \mid X = x, S_{jk} = 1)$ and (when unknown) $\pi_{jk}(x) = \mathbb{P}(S_{jk} = 1 \mid X = x)$ using global LightGBM classifiers that share parameters across items and contexts. Cross-fitting (Sec. 5) is used throughout; nuisance models are fit on the training folds and evaluated on the held-out fold.

## K.2. Feature construction

For each row corresponding to an observed ordered pair $(j, k)$ under context $x$:

$$\text{features} = \begin{bmatrix} x, \ j, \ k, \ (\text{optional judge features}) \end{bmatrix}.$$

Item indices $j$ and $k$ are treated as categorical integer features (instead of one-hot encoding). If a black-box judge provides class probabilities $\widehat{\mu}_{jk}^{\text{bb}}(x)$, we append the first $q = \max\{m - 1, 1\}$ components as additional numeric features for $\mu$.

## K.3. Training $\widehat{\mu}$

We train a multiclass LightGBM classifier on labeled pairs (i.e., rows with $S_{jk} = 1$), using the row-level design above. The model is always fit with the multiclass objective (even when $m = 2$) to produce well-formed class probabilities. Predictions are then produced for *all* ordered pairs $(j, k)$ for each context via batched inference, yielding $\widehat{\mu}(x) \in \mathbb{R}^{K \times K \times m}$. Diagonal entries are set to zero and off-diagonal rows are renormalized if needed. If only one class is observed in a training fold, we fall back to a constant predictor given by empirical class frequencies.

## K.4. Training $\widehat{\pi}$

When $\pi$ is unknown, we train a binary LightGBM classifier using the same features (without judge features). Positive examples are the observed ordered pairs in each context. Negatives are generated by sampling unobserved ordered pairs within each context at a fixed ratio `neg_per_pos`; sample weights correct for the negative subsampling rate. The model outputs $\widehat{\pi}_{jk}(x)$ for the requested rows. If only one class appears, we return the constant empirical rate.

## K.5. Hyperparameter tuning details

**Tuning procedure (shared).** We tune LightGBM classifiers with randomized search, using $K_{\text{CV}} = 3$ folds, $N_{\text{iter}} = 30$ draws, and score neg_log_loss. We run with $n_{\text{jobs}} = -1$ and use the same procedure for $\widehat{\mu}$ (and for $\widehat{\pi}$ when applicable).

**Tuning parameters (grid variables).** Let the tuned hyperparameters be

$$\Theta = \{M, \eta, L, n_{\min}, \rho, \rho_c, \lambda_1, \lambda_2, D\},$$

with the following meanings:

- $M$ (`n_estimators`): number of boosting iterations/trees.

- $\eta$ (`learning_rate`): shrinkage (step size).

- $L$ (`num_leaves`): maximum leaves per tree.

- $n_{\min}$ (`min_child_samples`): minimum samples per leaf.

- $\rho$ (`subsample`): row subsampling fraction per tree.

- $\rho_c$ (`colsample_bytree`): feature subsampling fraction per tree.

- $\lambda_1$ (`reg_alpha`): $\ell_1$ regularization on leaf weights.

---

[2]Code is available at https://github.com/DennisFrauen/NonparametricLLMEval.

- $\lambda_2$ (reg_lambda): $\ell_2$ regularization on leaf weights.

- $D$ (max_depth): maximum tree depth ($D = -1$ means no limit).

**Additional fixed tuning parameter.** For $\widehat{\pi}$, negative sampling uses neg_per_pos (number of negatives per positive); this is fixed per experiment (not tuned).

All experiments use randomized search with 3-fold CV, $n_{\text{iter}} = 30$, scoring neg_log_loss, and n_jobs $= -1$.

For the synthetic experiments, we use two distinct hyperparameter tuning grids shown in Table 3.

| Debiasing and BT misspecification experiments | Judge and data collection experiments |
|---|---|
| $n_{\text{estimators}} \in \{50, 100, 300, 600\}$ | $n_{\text{estimators}} \in \{100, 300, 600\}$ |
| $\eta \in \{0.03, 0.07, 0.1\}$ | $\eta \in \{0.03, 0.07, 0.1\}$ |
| $L \in \{15, 31, 63\}$ | $L \in \{15, 31, 63\}$ |
| $n_{\min} \in \{10, 30, 60, 80\}$ | $n_{\min} \in \{10, 30, 60\}$ |
| $\rho \in \{0.7, 0.9, 1.0\}$ | $\rho \in \{0.7, 0.9, 1.0\}$ |
| $\rho_c \in \{0.7, 0.9, 1.0\}$ | $\rho_c \in \{0.7, 0.9, 1.0\}$ |
| $\lambda_1 \in \{0.0, 0.1, 1.0\}$ | $\lambda_1 \in \{0.0, 0.1, 1.0\}$ |
| $\lambda_2 \in \{0.0, 0.1, 0.5, 1.0\}$ | $\lambda_2 \in \{0.0, 0.1, 1.0\}$ |
| $D \in \{-1, 6, 10, 15\}$ | $D \in \{-1, 6, 10\}$ |

*Table 3.* LightGBM tuning grids for synthetic experiments.

The tuning grid for real-world data is shown in Table 4.

| Hyperparameter | Search space |
|---|---|
| $n_{\text{estimators}}$ (n_estimators) | $\{50, 100, 300, 600\}$ |
| $\eta$ (learning_rate) | $\{0.03, 0.07, 0.1\}$ |
| $L$ (num_leaves) | $\{15, 31, 63\}$ |
| $n_{\min}$ (min_child_samples) | $\{10, 30, 60, 80\}$ |
| $\rho$ (subsample) | $\{0.7, 0.9, 1.0\}$ |
| $\rho_c$ (colsample_bytree) | $\{0.7, 0.9, 1.0\}$ |
| $\lambda_1$ (reg_alpha) | $\{0.0, 0.1, 1.0\}$ |
| $\lambda_2$ (reg_lambda) | $\{0.0, 0.1, 0.5, 1.0\}$ |
| $D$ (max_depth) | $\{-1, 6, 10, 15\}$ |

*Table 4.* LightGBM hyperparameter search space used for real-world data experiments.

# L. Details regarding synthetic data

In this section we provide details regarding synthetic data generation in our experiments from Sec. 7[3].

## L.1. Common synthetic data generation pipeline

**Contexts and selection.** For each dataset, contexts are sampled as $X_i \sim \mathrm{Unif}([0,1]^p)$ for $i = 1, \ldots, n_{\mathrm{ctx}}$. For each ordered pair $(j, k)$, we draw $S_{i,jk} \sim \mathrm{Bernoulli}(\pi_{jk}(X_i))$ with $S_{i,jj} = 0$, and keep only rows with $S_{i,jk} = 1$.

**Outcomes.** Given selected $(j, k)$, we draw $Y_{i,jk} \sim \mu_{jk}(X_i)$, either Bernoulli (binary) or categorical (one-hot). Ground-truth GRE scores are computed by averaging $F(\mu(X))$ over $10^6$ fresh contexts.

## L.2. Simulators used (DGPs)

We use two simulators to generate synthetic preference data for our experiments.

### L.2.1. NONLINEAR TIES SIMULATOR

Three categories: win/lose/tie.

For item $j$ we define the utilities

$$u_j(x) = W_j^\top x + b_j + W_j^{(q)\top} x_{1:q}^{\odot 2} + s \cdot \sin(2\pi x_0 + \phi_j).$$

with default values $\sigma_\ell{=}1$ (scale of $W_j$), $\sigma_q{=}0.6$, $s{=}0.6$, $q = \min(2, p)$.

**Outcome model.** Define

$$d_{jk}(x) = \frac{u_j(x) - u_k(x)}{T} + \beta\,(x_0 - 0.5), \qquad \ell_{jk}^{\mathrm{tie}}(x) = \tau_0 - \tau_1 |d_{jk}(x)| + \tau_c \cdot \cos(2\pi x_{1 \text{ or } 0}),$$

and

$$\mu_{jk}(x) = \mathrm{softmax}\big([d_{jk}(x), -d_{jk}(x), \ell_{jk}^{\mathrm{tie}}(x)]\big).$$

Here, we set $T{=}1$, $\beta{=}0$, $\tau_0{=}0.2$, $\tau_1{=}1.2$, $\tau_c{=}0.4$. A minimum mass $\varepsilon_\mu$ is enforced (configurable; see Table 5).

**Selection model.** For a base rate $\pi_p$ we define

$$\pi_{jk}(x) = (1 - \lambda_\pi)\pi_p + \lambda_\pi\, \sigma\big(\mathrm{logit}(\pi_p) - \kappa_\pi |d_{jk}(x)| + \eta_\pi(x_0 - 0.5) + b_j^\pi + b_k^\pi\big),$$

with defaults $\kappa_\pi{=}0.8$, $\eta_\pi{=}0.4$, $\pi_{\min}{=}0.01$, $\pi_{\max}{=}0.5$. (Configurations override $\lambda_\pi$, $\pi_{\min}$, $\pi_{\max}$ as needed.)

### L.2.2. BT MISSPECIFICATION SIMULATOR

Two categories: win/lose.

**Outcome model.** With the same $u_j(x)$ as above,

$$d_{jk}(x) = \frac{u_j(x) - u_k(x)}{T} + \beta(x_0 - 0.5), \qquad \mu_{jk}(x) = \sigma(d_{jk}(x) + \gamma\, C_{jk}),$$

where $C$ is the cycle matrix and $\gamma$ controls BT misspecification. Defaults: $T{=}1$, $\beta{=}0$, $\gamma$ is set per experiment.

**Selection model.** Same $\pi_{jk}(x)$ as above with the same defaults.

## L.3. Synthetic experiment data configurations

Table 5 reports the data parameters used in all synthetic experiments from Sec. 7.

**Judge noise.** In `synthetic_ties_judge`, the judge is constructed by adding Gaussian noise on the logit scale of the ground-truth $\mu$ with $\sigma_{\mathrm{judge}} \in \{0, 0.2, 0.4, 0.6, 0.8, 1\}$.

---

[3]Code is available at https://github.com/DennisFrauen/NonparametricLLMEval.

| Setup | Simulator | $K$ | $p$ | $n_{\text{ctx}}$ | Data parameters |
|---|---|---|---|---|---|
| Effectiveness of debiasing | NonlinearTie | 3 | 2 | sweeping | $\pi_p{=}0.3, \lambda_\pi{=}0.1, \varepsilon_\mu{=}0.05, \pi_{\min}{=}0.05, \pi_{\max}{=}0.5$ |
| External synthetic judges | NonlinearTie | 3 | 5 | 2000 | $\pi_p{=}0.3, \varepsilon_\mu{=}0.05, \pi_{\min}{=}0.05$ |
| Optimal data collection | BTMisspec | 3 | 2 | 3000 | $\pi_p{=}0.3, \lambda_\pi{=}0.1, \gamma{=}1$ |
| BT-model misspecification | BTMisspec | 4 | 5 | 2000 | $\pi_p{=}0.3, \lambda_\pi{=}0.1, \gamma \in \{0, 0.5, 1, \ldots, 4\}$ |

*Table 5.* Synthetic data configurations.

# M. Details regarding real-world data

In this section we provide details regarding real-world datasets used in Sec. 7 and Appendix N.

## M.1. Datasets and variables

We evaluate on two public real-world preference datasets, both originally provided by Zheng et al. (2023).

**Chatbot Arena conversations.** Each row contains a prompt (up to three turns), a model pair $(\text{model\_a}, \text{model\_b})$, and a categorical winner label in $\{\texttt{model\_a}, \texttt{model\_b}, \texttt{tie}, \texttt{tie (bothbad)}\}$. We deduplicate by `question_id` and use one comparison per context, yielding $n = 32{,}980$ contexts/comparisons and $K = 20$ distinct models.

**MT-Bench human judgments.** We compute a majority-vote winner for each $(\texttt{question\_id}, \texttt{turn}, \texttt{model\_a}, \texttt{model\_b})$ over human judges. This yields $n = 2{,}396$ pairwise comparisons across 160 unique contexts, with labels in $\{\texttt{model\_a}, \texttt{model\_b}, \texttt{tie}\}$ and $K = 6$ models.

**Items and labels.** In both datasets, models are mapped to integer IDs and used as items. Labels are one-hot encoded with $C = 4$ categories for Chatbot Arena and $C = 3$ for MT-Bench.

| Dataset | $K$ | $C$ | #contexts | #comparisons | Features ($p$) |
|---|---|---|---|---|---|
| Chatbot Arena | 20 | 4 | 32,980 | 32,980 | 102 |
| MT-Bench (human) | 6 | 3 | 160 | 2,396 | 101 |

*Table 6.* Real-world datasets used in experiments.

## M.2. Preprocessing and text representation

We obtain prompt representations vectorized by TF-IDF, then reduced with truncated SVD (PCA) to 100 dimensions. For Chatbot Arena, we concatenate: (i) a RoBERTa-based toxicity probability, (ii) the 100-D SVD prompt vector, and (iii) the turn index, giving $p = 102$. For MT-Bench, we concatenate the 100-D SVD prompt vector and the turn index, giving $p = 101$.

## M.3. Experimental configuration

We run a single experiment per dataset (`n_runs=1`) and use cross-fitting with $V = 2$ folds for Chatbot Arena and $V = 5$ folds for MT-Bench. We report both debiased (AIPTW) and plug-in estimators for Borda, BT, and rank centrality scores, and compute simultaneous CIs using `rect` and `bonferroni` with $\alpha = 0.05$.

## M.4. Nuisance training details

We train LightGBM classifiers for $\widehat{\mu}$ and $\widehat{\pi}$. Tuning uses randomized search with 3-fold CV and the `neg_log_loss` score. For Chatbot Arena, we use $N_{\text{iter}} = 2$; for MT-Bench, we use $N_{\text{iter}} = 30$. Negative sampling for $\widehat{\pi}$ uses `neg_per_pos`=10.

# N. Additional experiments

## N.1. Additional real-world dataset

Here, we report the additional real-world data results using the MT-Bench dataset introduced in Appendix M. The results are shown in Fig. 5.

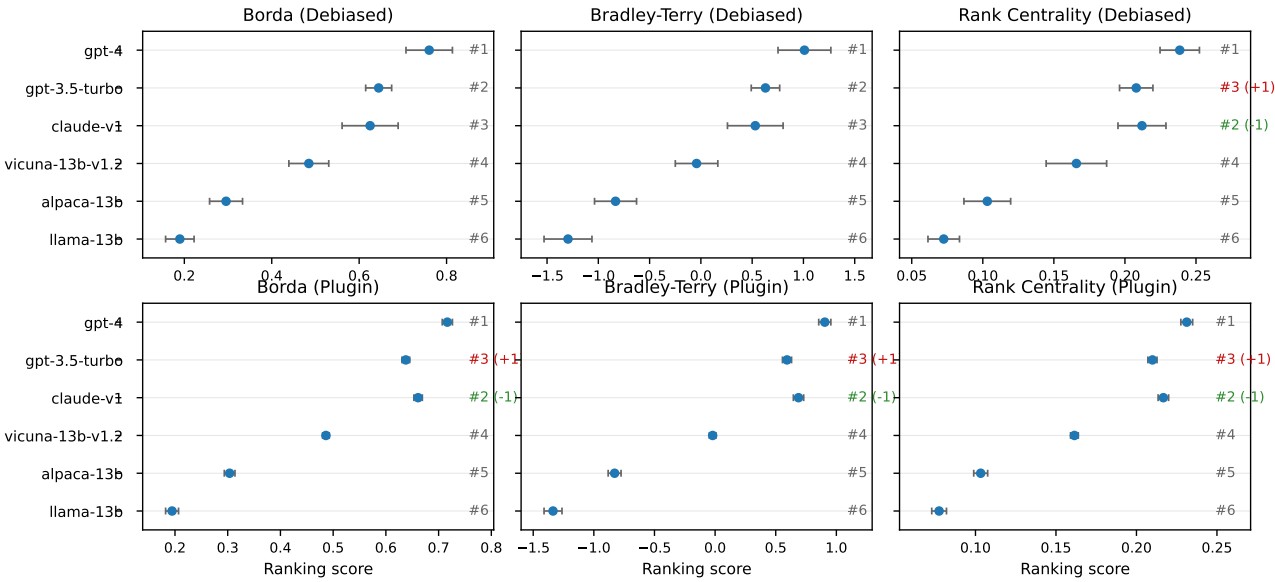

*Figure 5.* **Results for MT-Bench preference data.** Shown: estimated ranking scores for different GARS functionals (Borda, BT, and rank centrality) and estimators (debiased and plug-in). Changes in ranking are indicated in red and green (for debiased estimators as compared to debiased Borda scores, for plug-in estimators as compared to the corresponding debiased estimator). We report 95% simultaneous confidence intervals, which attain near-zero width for plug-in estimators.

**Results.** The results are shown in Fig. 5. We make similar observations as for the real-world experiment in Sec. 7: (i) all obtained rankings are largely consistent with the baseline ranking, thus indicating a reasonable base performance. (ii) Rank Centrality deviates slightly more from the baseline than Borda and RC, thus highlighting that different GARS types can yield different rankings even on the same data. (iii) While the point ranking scores of the plug-in estimators do not deviate too much from their debiased counterparts, their confidence intervals are very small and are most likely invalid.

## N.2. Real-world results with OpenAI embeddings

Here we show the results for Chatbot Arena preference data from Figure 3 of the main paper but with OpenAI text embeddings. We report 95% simultaneous confidence intervals, which attain near-zero width for plug-in estimators. Prompt representations are obtained using OpenAI text embeddings (`text-embedding-3-small`[4]) with dimensionality set to 100, replacing the TF-IDF+SVD representation with a semantic encoding of prompt context. **The overall results remain robust.**

---

[4]https://platform.openai.com/docs/guides/embeddings

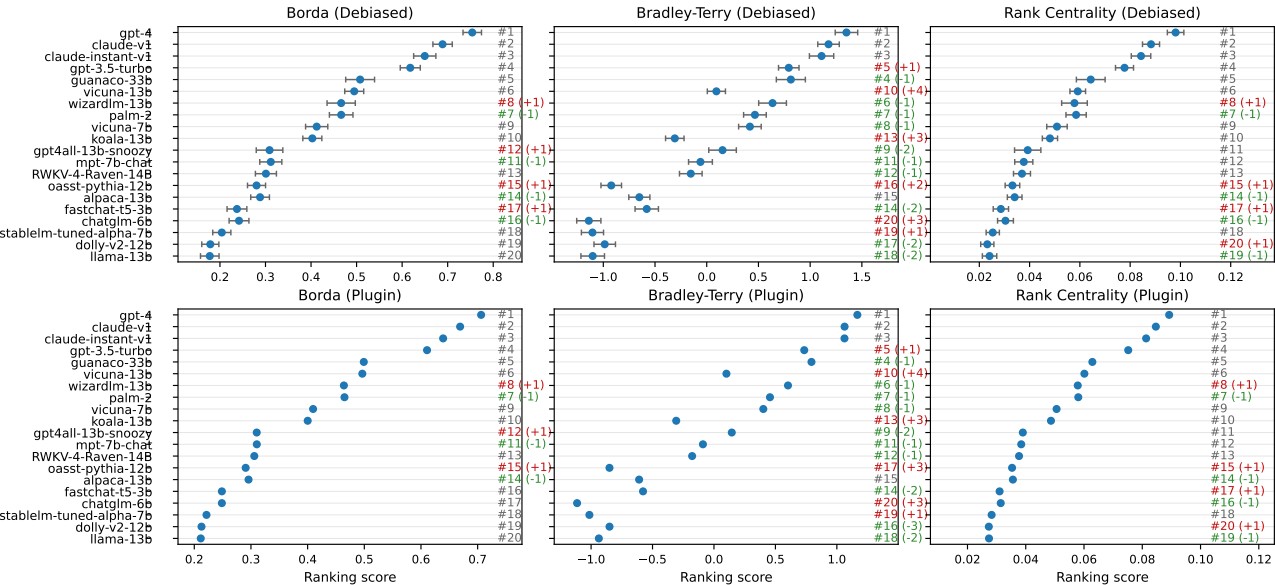

*Figure 6.* **Results for Chatbot Arena preference data from Figure 3 of the main paper but with OpenAI text embeddings.** Shown: estimated ranking scores for different GARS functionals (Borda, BT, and rank centrality) and estimators (debiased and plug-in). Changes in ranking are indicated in red and green as compared to the baseline ranking from Zheng et al. (2023).

