# OpenReview forum: "Nonparametric LLM Evaluation from Preference Data"
_ICML.cc/2026/Conference — ICML 2026 regular_

### Official Review · Reviewer_tpeM · 2026-03-12

**Soundness:** 4
**Presentation:** 3
**Significance:** 3
**Originality:** 3
**Overall Recommendation:** 5
**Confidence:** 4

**Summary:**

The paper introduces a framework (DMLRank) for ranking items (LLMs but possibly other types) based on pairwise preference data. The first contributions is a class of estimands that include multiple ranking methods (Bradley-Terry, Borda scores, etc.) as a choice of functional on the preference distribution. The second is an efficient DML-based estimator for this estimand with an asymptotic distribution in closed form, allowing computation of asymptotic confidence intervals. The last is a data gathering policy that is optimal for some metric based on this asymptotic form. Experiments show the benefits of the efficient estimator and calibration of its confidence intervals compared to the naive plug-in estimator.

**Compliance With Llm Reviewing Policy:**

Affirmed.

**Final Justification:**

I keep my Accept score. I appreciate the authors' transparency and engagement in the rebuttal process. A number of points discussed in that process will help clarify and strengthen the contribution, which I think is solid and impactful both in terms of theoretical unification and practical estimation efficiency.

**Key Questions For Authors:**

1. Were the forms (3), (5) and (6) previously known?
2. Did you observe more significant improvements from the optimal policy than shown in Table 2?
3. In what way does the (non-)linearity of $F$ affects the theoretical or practical properties of the estimator?

**Limitations:**

yes

**Strengths And Weaknesses:**

**Soundness** The paper is technically sound. The formalism is solid and extensive derivations are provided in the appendix. The efficiency, calibration, and robustness of the estimate is well justified by the experiments. The results on the optimal policy are not significant.

**Presentation** The paper is well-structured overall. It does a lot, which makes it rather cramped for the 8-page format, leading for example to no conclusion. Perhaps due to conciseness it is not always clear what is a contribution of the paper. For example the formulations in (3) and (6) to cast the respective methods to the GARS form. On the former, the exposition is rather rushed and would benefit from an expanded explanation in the appendix for the readers less versed in graph theory. At a higher level, DML is standard (and originated) in the causal inference literature. A more direct connection could have made the exposition more intuitive and isolate the contribution of this paper. My attempt at this follows.

In causal inference, the average treatment effect is $\tau = \mathbb{E}[Y^{(1)} - Y^{(0)}]$. The simple Borda scores can be written as $\theta_j = \mathbb{E}[\frac{1}{2(K-1)}\sum_{k\neq j}(Y_{jk1} + Y_{kj2})]$. In both cases we do not observe all components of $Y$ and we are only interested in the expectation of a linear functional of it, not its full distribution. The formulation in (4) somewhat obscures this connection. If $F$ is linear (as in the Borda case), then $\mathbb{E}[F(\mu(X))] = \mathbb{E}[F(Y)]$. Otherwise, we can recover a similar linear form at first order around an estimate $\hat\mu$ to get $\mathbb{E}[F(\mu(X))] \simeq \mathbb{E}[F(\hat\mu(X))] + \mathbb{E}[J(\hat\mu(X))(Y - \hat\mu(X))]$. The link is then clear with the causal setting and similar methods can apply.

**Significance** While LLM leaderboards can be of controversial utility, the applications of this work go beyond that, including any kind of ranking from preference data.

**Originality** The paper provides useful unifying insights between different ranking methods. The unified form being a contribution, it would have been interesting if the author used this insight to propose new ranking methods. I also defer to questions to better understand the contribution in that unification.

---

> ### Author Rebuttal · Authors · 2026-03-30
>
> Thank you for your positive review and your helpful comments. Below we have drafted careful responses to your suggestions. We will incorporate all points marked with **Action** into the revised version of our paper.
>
> ## Response to Presentation
>
> * **Missing conclusion / paper organization.** Thank you for pointing this out. **Action:** In the revised version, we will add a concluding discussion to the main paper that clearly summarizes the contributions, limitations, and future work directions.
> * **Eqs. (3), (5), and (6) (GARS examples).** The underlying ranking ideas themselves are largely established in the literature: e.g., Borda-type average win-rate scores, and Rank Centrality / PageRank-style stationary-distribution scores are well known. The BT projection in Eq. (5) can be viewed as an $L_2$-type projection of the contextual preference probabilities onto the BT score class.
>
>     However, **what is new in our paper** is not the individual ranking scores per se, but the **explicit formulation of these ranking targets as averages of functionals of the underlying preference function $\mu$**, i.e., as $\theta=\mathbb E[F(\mu(X))]$. This formulation turns a range of ranking methods into a **single statistical estimation problem with nuisance functions**, which is precisely what makes a unified DML approach possible.
>
> * **Connection to DML in causal inference.** Thank you also for providing this intuition; we found it very helpful and agree that making this connection more direct would improve the exposition. We will add a paragraph to the revised version of our paper.
>
>     One caveat is that **DML in causal inference applies after making identifiability assumptions** that rewrite targets involving unobserved potential outcomes into functionals of observed-data quantities. For example, the ATE can be written as $\mathbb E[\mu_1(X)-\mu_0(X)], \qquad \mu_a(X)=\mathbb E[Y\mid X,A=a]$. For Borda, one can write a population target in terms of potentially unobserved pairwise outcomes. Under our identifiability assumptions (in particular, missing at random conditional on $X$), this target becomes the identified observed-data quantity in Eq. (3), namely a functional of $\mu(X)$. Thus, Eq. (3) is the analogue of the identified causal target that no longer depends on unobserved preferences. That said, our contributions to derive DML methodology for generalized ranking estimands is consistent with a large body of established works that derive DML methods for specific statistical inference settings (see refs in our other rebuttals) and require non-trivial modifications (derivation of the EIF, establishing asymptotic normality).
>
>
> ## Response to Originality
>
> * **New ranking estimands compatible with GARS.** Indeed, our unified GARS form can also be used to define *new* ranking estimands via a suitable choice of $F$. In the following, we provide two examples:
> 1. For binary $\mathcal{C}=\{1,2\}$, we can define a score that rewards only sufficiently confident wins: $F_j(\mu(x))=\frac{1}{2(K-1)}\sum_{k\neq j}\Big(\phi_{\tau}(\mu_{jk1}(x))+\phi_{\tau}(\mu_{kj2}(x))\Big)$, where $\phi_{\tau}:[0,1]\to[0,1]$ is e.g., $\phi_{\tau}(u)=\mathbf{1}\{u\geq \tau\} \qquad\text{or}\qquad \phi_{\tau}(u)=\sigma\big(\alpha(u-\tau)\big)$ with threshold $\tau > 1/2$ and sharpness parameter $\alpha>0$. This score emphasizes *clear* pairwise wins rather than marginal ones.
> 2. A *robust / worst-case score*, which favors items that do not have a pronounced weakness against any competitor: $F_j(\mu(x))=\min_{k\neq j}\frac{\mu_{jk1}(x)+\mu_{kj2}(x)}{2}$ (or alternatively a smoothed version).
>
>     **Action:** We will add these examples to the paper.
>
>
>
> ## Response to Questions
>
> 1. To avoid redundancies, we kindly point to our “Response to Presentation”.
> 2. The results for data-collection are quite noisy because each run involves resampling of the dataset according to the data-collection policy and the preference probabilities, and each run is computationally expensive since we re-tune and re-train the nuisance models. Nevertheless, we re-ran the experiment with 50 runs [here](https://anonymous.4open.science/r/NonparametricLLMEval-603E/doc/DMLRank_rebuttal.pdf).
> 3. Our framework and all main theoretical results apply to *both* linear and nonlinear $F$. That said, linearity brings two specific simplifications/ advantages:
>    1. **Simpler structure.** If $F$ is linear, the Jacobian is constant, and some regularity conditions used in the appendix become immediate (e.g., Lipschitz-type smoothness conditions are trivial).
>    2. **Double robustness.** For linear $F$, one can additionally obtain a double-robust property: the estimator remains consistent if either the preference model or the propensity model is correctly specified (similarly to doubly robustness of the AIPTW estimator in causal inference). In general, this property is lost for nonlinear $F$, where correct estimation of $\mu$ becomes essential (see Assumption A4).

---

> > ### Author Rebuttal · Reviewer_tpeM · 2026-04-01
> >
> > I appreciate the thoughtful rebuttal. The new ranking estimands are interesting and will surely strengthen the paper. Discussion of the linear case and the double robustness property could also be of interest to the reader as at least one of the estimands has linear $F$, and help highlight what is lost by its non-linearity. I only seek confirmation on the first point.
> >
> > 1. On Eqs. (3), (5), and (6) my question was whether those specific forms were known, not the underlying scoring method. Most particularly, going from (1) to (3) is non-trivial and I wanted to confirm whether that was a contribution of this paper. The paper stresses the novelty of the general form but not necessarily of the work to convert known methods to that form.
> >
> > 2. On the link to causal inference, I am simply talking about intuition, which of course will vary with individuals so this is not a counterargument. I find writing the target as a function of $Y$, as I attempted in my review, clearer than writing it as a function of $\mu(X)$, since $Y$ is the observable. It highlights that the estimation challenge, just as in causal inference, is that $Y$ is not observed for all potential outcome/pairs. It also directly links to the discussion on the linearity of $F$. Writing the estimand as a function of $\mu(X)$ does not immediately mean that it doesn't depend on unobserved preferences, $\mu(X)$ is not observed either. However, I can understand that this links more directly to the identifiability assumptions. An estimand like $\mathbb{E}[Y_{ij}Y_{kl}]$ is a clear function of $Y$ but may have different identifiability assumptions.

---

> > > ### Author Response · Authors · 2026-04-02
> > >
> > > Thank you for your engagement in the rebuttal and your thoughtful follow-up. We will add the points mentioned (particularly the new ranking estimands and discussion on linearity of $F$) to the camera-ready version of our paper. We respond to the two remaining points below.
> > >
> > > ## Point 1
> > >
> > > **Going from Eq. (1) to (3).** Thank you for pointing out the ambiguity here. **Action:** We will revise the appendix to make the step from Eq. (1) to Eq. (3) explicit. Under the BT model, it holds that $\mu_{jk1}(x)=\sigma(r_j(x)-r_k(x)+b(x))$ and using the reversed ordering, $\mu_{kj2}(x)=\sigma(r_j(x)-r_k(x)-b(x))$. Therefore, the symmetrized log-odds satisfy
> > >
> > > $\ell(\mu(x)) = \left(\frac{logit(\mu_{jk1}(x))+logit(\mu_{kj2}(x))}{2}\right)_{j<k}$
> > >
> > > $= (r_j(x)-r_k(x))_{j<k} = Br(x)$.
> > >
> > > Hence, under BT, Eq.~(3) solves
> > >
> > > $\min_{\phi:\,\mathbf{1}^\top \phi = 0}\|B\phi-\ell(\mu(x))\|_2^2$
> > >
> > > $=\min_{\phi:\,\mathbf{1}^\top \phi = 0}\|B\phi-Br(x)\|_2^2$.
> > >
> > > Since $r(x)$ is feasible and achieves objective value $0$, it is the minimizer; with the zero-sum normalization, it is unique. Thus, Eq.~(3) exactly recovers the BT score vector when the BT model holds.
> > >
> > > **Novelty of Eq.(3), (5), and (6)**. Our main reason for introducing Eq.~(3) was to obtain a *closed-form BT-type projection* written directly in terms of the preference probabilities $\mu$ (for use within the GARS framework. We are not aware of this exact formulation in the literature, but we do not want to overclaim novelty: related projection-based formulations exist, e.g., in the HodgeRank literature [1], where authors project an observed pairwise comparison flow $Y$ onto the score-difference / gradient subspace. In contrast, our formulation applies the projection to the *symmetrized log-odds* $\ell(\mu(x))$ and does so in the *contextual, nonparametric* setting.
> > >
> > > This is similar for Eq. (5) (Borda) and (6) (Rank-Centrality). The underlying notions have been proposed in classical, non-contextual settings (e.g., [2] for Borda, [3] for Rank-Centrality), but to our knowledge have not yet been explicitly written down in our specific setting.
> > >
> > > *Takeaway: Our contributions for Eq. (3)-(6) can be thought of as generalizing established notions to our contextual and nonparametric preference evaluation setting as well as proposing a unifying formulation (GARS) for statistical inference.*
> > >
> > > **Action:** We will add the above clarifications to our camera-ready version.
> > >
> > >
> > > ## Point 2
> > >
> > > Thank you for elaborating. We agree that writing the estimands in terms of potential/ unobserved outcomes can help to provide intuition and to compare estimation challenges with classical problems such as ATE estimation in causal inference (particularly, since we also use similar identifiability assumptions such as missing at random). In our camera-ready version, we propose to add both a paragraph on this (i.e., including your considerations above and from the initial review) as well as a discussion on identifiability assumptions and results to make the connection to causal inference clean. We remain open for feedback or ideas.
> > >
> > > Thank you again for your engagement and helpful suggestions!
> > >
> > > ## References
> > >
> > > [1] Jiang et al. (2010). Statistical Ranking and Combinatorial Hodge Theory. Mathematical Programming.
> > > [2] Shah et al. (2018). Simple, Robust and Optimal Ranking from Pairwise Comparisons. JMLR.
> > > [3] Negahban et al. (2017). Ranking from pair-wise comparisons. Operations Research.

---

### Official Review · Reviewer_Ydy6 · 2026-03-13

**Soundness:** 3
**Presentation:** 3
**Significance:** 3
**Originality:** 3
**Overall Recommendation:** 5
**Confidence:** 3

**Summary:**

This paper addresses the statistical challenges encountered when LLMs evaluate performance using paired preference data. Compared to parameterized models that rely on model setting biases, the authors propose a non-parametric statistical framework for debiased machine learning called GARS. This framework unifies multiple ranking metrics by introducing generalized average ranking scores. The authors derived the effective influence function of GARS and established the asymptotic normality of the estimator under weaker conditions. They also conducted evaluations on synthetic data and real-world datasets.

**Compliance With Llm Reviewing Policy:**

Affirmed.

**Key Questions For Authors:**

In Table 1, there is a significant gap between the virtual data and the plugin method. However, in Fig 3, although the debiased method has larger confidence intervals compared to the plugin method, there is no significant difference in ranking.

**Limitations:**

Yes

**Strengths And Weaknesses:**

Strengths

This work is relatively easy to understand. When introducing flexible neural network methods to predict preferences, it is difficult to obtain effective statistical uncertainty quantification. The authors provide a non-parametric and more reliable solution. The paper also mentions an interesting location bias phenomenon and gives corresponding solutions.

Weaknesses

1. The text in Figures 1 and 2 is too small.
2. Clarify the key differences from the existing debiasing methods.

---

> ### Author Rebuttal · Authors · 2026-03-30
>
> Thank you very much for your thoughtful review! Below, we address all your comments and suggestions. We will incorporate all points marked with **Action** into the revised version of our paper.
>
> ## Response to Weaknesses
>
>
> 1. **Text size in Figures 1 and 2.** Thank you for pointing this out. **Action:** We will increase the font size and improve the readability of both figures in the revised version.
> 2. **Key differences from existing DML methods.** Debiased / double machine learning (DML) is a *general* methodological framework for efficient estimation and valid inference when the target parameter depends on high-dimensional nuisance functions estimated with ML (see Rubin, van der Vaart, Bickel et al., Chernozhukov et al.). In our setting, the target parameter is the GARS vector $\theta$, while the nuisance parameters are the preference functions $\mu$ and propensities $\pi$. A naive plug-in approach first estimates $\mu$ with ML and then substitutes $\hat\mu$​ into the ranking functional, but this can induce plug-in bias and invalid confidence intervals. Our DML approach addresses this by using the efficient influence function (EIF), which corrects this bias and yields statistically efficient estimation and valid confidence intervals under weaker assumptions.
>
>     That said, **our contribution is not the general DML idea itself, but its new derivation and instantiation for generalized ranking estimands from preference data which is non-trivial.** Here, our contributions are consistent with a large body of established works that derive DML methods for specific statistical inference settings [e.g., 1, 2, 3]. To the best of our knowledge, **no prior work has derived a DML estimator for generalized ranking estimands in the form we study here.** Our technical contributions include:  (i) introducing GARS as a unified nonparametric class of ranking estimands; (ii) deriving the EIF for GARS and proving efficiency and asymptotic normality (Theorem 1); and (iii) deriving the optimal data-collection policy for these estimands (Theorem 2).
>
>
>     **Compared to existing debiasing methods, our framework is thus specific to preference-based ranking and provides a unified EIF-based treatment for multiple ranking notions such as Borda, BT-type scores, and rank centrality.**
>
>
>     **Action:** We will revise the paper to better distinguish between the general DML framework and our novel derivations/contributions in the GARS setting.
>
>
>
> ## Response to Question
>
> **Why Table 1 shows a large gap, while Fig. 3 shows less difference in ranking.** Thank you for this important question. The two results highlight different aspects of the method. Table 1 is based on synthetic data designed to exhibit substantial plug-in bias, so the difference between debiased and plug-in estimators is intentionally pronounced there; this allows us to directly verify the theoretical claims on bias reduction, efficiency, and valid coverage. In contrast, Fig. 3 uses a real-world dataset, where the magnitude of plug-in bias is data-dependent and can be smaller. Thus, the visual gap in point estimates can appear less dramatic.
>
> At the same time, **Fig. 3 still shows meaningful differences:** the scores are not identical, and they do lead to large ranking changes, as indicated by the red/green ranking markers in the bottom row. More importantly, the main issue in Fig. 3 is the uncertainty quantification: the plug-in confidence intervals are artificially narrow, whereas the debiased intervals account for first-stage estimation error and are therefore valid. So even when the ranking differences are smaller in a given dataset, the advantage of debiasing remains that it provides principled inference and avoids misleadingly overconfident CIs. Also, if plug-in bias happens to be small in a specific dataset, DML typically does not hurt performance, while still preserving valid inference. \
> **Action:** We will clarify in the revision that the different roles of Table 1 and Fig. 3 (which is expected because plug-in bias depends on the data-generating process), and that Fig. 3 primarily illustrates the validity of uncertainty quantification in real data.
>
>
> ## References
>
> [1] Kallus et al. (2022). What's the Harm? Sharp Bounds on the Fraction Negatively Affected by Treatment. NeurIPS.
>
> [2] Dorn et al. (2025). Doubly-Valid/Doubly-Sharp Sensitivity Analysis for Causal Inference with Unmeasured Confounding. Journal of the American Statistical Association.
>
> [3] Kennedy et al. (2023). Towards optimal doubly robust estimation of heterogeneous causal effects. Electronic Journal of Statistics.

---

> > ### Author Rebuttal · Reviewer_Ydy6 · 2026-04-03
> >
> > The authors provide clear responses to my concerns. The clarification of the novelty relative to existing methods is helpful, and the discussion on Table 1 vs. Fig. 3 explains my questions. I maintain my original score.

---

> > > ### Author Response · Authors · 2026-04-03
> > >
> > > Thank you! We sincerely appreciate your time and effort in evaluating our paper and will incorporate all action points to the camera-ready version.

---

### Official Review · Reviewer_AJjT · 2026-03-24

**Soundness:** 4
**Presentation:** 3
**Significance:** 3
**Originality:** 3
**Overall Recommendation:** 5
**Confidence:** 4

**Summary:**

When we rank LLMs on say Chatbot Arena or LMSys, we use BT which then calculates ELOs based on win/loss ratios. The paper says we should not do this for two main reasons. BT assumes a specific curve for win rate translating to scores. Furthermore it can contain ranking loops. In such cases the BT forces a ranking which is biased. Furthermore if ML models are used to rank in LMSys etc, the CI's are not trustworthy. To mitigate this effect they use a non-parametric framework for mimic any ranking system like Brda/Pagerank etc.

An important contribution of the framework is telling us exactly which two LLMs should you compare to shrink the error bars. So this can be used to determine what should be shown in LMSys next.

**Compliance With Llm Reviewing Policy:**

Affirmed.

**Key Questions For Authors:**

1. What is Assumption A5? (typo?)

**Strengths And Weaknesses:**

## Strengths

1. Studies a solid research problem which is important for the community. Ranking can be cyclic or have non monotone order. How do we use
2. GARS as the estimand and DMS as the estimator class is pretty solid. An aspect i really liked is the methodological narrative.They start from preference probabilities, then showwhy plug-in estimation is biased for nonlinear ranking functionals, and then build the debiased correction, derived from the derivative/Jacobian of the ranking map. While this was a hard read, I thought this was well thought through.
3. The experiments while weak directly address the theoretical claims. The theoretical analysis is also strong, and the paper is a good fit for ICML.


## Weaknesses
1. The paper's experiments while demonstrative are not sufficient for a full scale LLM evaluation. It would be great if they could run a simulation, if possible over some dataset of their choice?
2. TF-IDF and SVDs in the current era are toy examples, no?
3. Are the assumptions too strong? Can you comment on Y independent of S given X (Missing at random assumption)?

---

> ### Author Rebuttal · Authors · 2026-03-30
>
> Thank you very much for your positive review! Below, we address your comments and suggestions in detail. We will incorporate all points marked with **Action** into the revised version of our paper.
>
>
> ## Response to Weaknesses
>
>
>
> 1. **Scope of the experiments / “full-scale” LLM evaluation.** Our goal in this paper is primarily methodological: to introduce DMLRank and empirically verify that it behaves as predicted by the theory. For that reason, the main experiments are based on synthetic data, where ground-truth ranking scores are available and where we can directly evaluate estimation error and confidence interval coverage. In contrast, for real-world LLM preference datasets, there is no ground-truth ranking, so one cannot rigorously assess whether a method recovers the “correct” ranking. Thus, consistent with the DML literature [e.g., 1, 2, 3], we use synthetic data for controlled validation and real-world data to demonstrate applicability. **Our goal is therefore not to provide a state-of-the-art leaderboard of the newest LLMs, but to introduce and rigorously validate a general methodology.**
> 2. **TF-IDF / SVD in the real-world experiment.** We agree that TF-IDF/SVD are not meant to be an optimal modern representation. In the real-world experiment, they are a deliberately simple instantiation chosen for simplicity and for obtaining a low-dimensional representation that can be used for downstream regression without exploding the feature space (given limited sample size). **Importantly, the DMLRank framework itself is model-agnostic and can be combined with richer text representations and stronger predictors**, e.g., learned embeddings or end-to-end text models. Our aim in this experiment was to demonstrate how DMLRank can be applied in practice and that it yields a reasonable ranking, **rather than to claim that this particular feature pipeline is optimal.**
>
>    **Action:** We agree that an additional ablation with state-off-the-art text embeddings will help to strengthen our paper. Hence, **we performed additional experiments** and re-ran our experiments from Fig. 3 (LMArena preference data) **using OpenAI text embeddings** (text-embedding-3-small). The results are shown in the following linked PDF and remain largely robust: [https://anonymous.4open.science/r/NonparametricLLMEval-603E/doc/DMLRank_rebuttal.pdf](https://anonymous.4open.science/r/NonparametricLLMEval-603E/doc/DMLRank_rebuttal.pdf).
> 3. **Assumptions / missing at random.** Thank you for raising this important point. We agree that the missing-at-random assumption (MAR) can be violated in practice. We would like to emphasize two points. (i) The richer the covariates $X$, the more plausible MAR becomes; in practice, one can include additional context such as prompt characteristics, evaluator metadata, or rater information whenever available. (ii) If MAR is violated, then the ranking target is not identifiable from observed preference data alone, so **any method based on such data (not only ours) but also established approaches such as BT-style leaderboard estimation can be biased.** We make this assumption explicit for transparency. Extending the framework to settings with MAR violations, e.g., via sensitivity analysis or partial identification ideas from causal inference, could be an important direction for future work.
>
> **Action.** We will revise the paper to better clarify: (i) that the main experimental validation is intentionally simulation-based because ground-truth rankings are needed, (ii) that the real-world study is a demonstration of applicability rather than an optimized leaderboard pipeline, and (iii) limitations of the MAR assumption and how to address this in future work.
>
>
> ## Response to Question
>
> **Assumption A5.** Thank you for catching this typo. We were referring to Assumption A4, i.e., the nuisance-rate assumptions used for the DML result, and we will correct this in the revised version.
>
>
>
>
> ## References
>
> [1] Curth et al. (2021). Nonparametric estimation of heterogeneous treatment effects: From theory to learning algorithms. AISTATS.
>
> [2] Chernozhukov et al. (2018) Double/De-Biased Machine Learning for Treatment and Causal Parameters. Econometrica.
>
> [3] Van der Laan. (2006). Targeted Maximum Likelihood Learning. The International Journal of Biostatistics.

---

> > ### Author Rebuttal · Reviewer_AJjT · 2026-04-02
> >
> > This paper is heading towards acceptance given the current scores. But for better citability, i recommend the authors consider larger scale experiments.

---

> > > ### Author Response · Authors · 2026-04-03
> > >
> > > Thank you for this helpful suggestion and for your positive assessment of the paper. In the camera-ready version, we will incorporate the additional experiments using OpenAI embeddings provided in the rebuttal and we are confident that this will increase the credibility of our real-world experiments. We would also like to point out that Appendix L already contains additional experiments in which we apply our ranking methodology to MT-Bench data, which is another well-established benchmark alongside Chatbot Arena.
> > >
> > > That said, we would like to emphasize that the primary goal of this paper is not to provide state-of-the-art benchmarks for LLMs. Rather, our main contribution is to propose and validate a new methodology, with a particular focus on synthetic data where the ground truth is known and the properties of the method can be studied in a controlled way.
> > >
> > > Thank you again, we sincerely appreciate your time and effort in reviewing our paper.

---

### Official Review · Reviewer_1SSE · 2026-03-25

**Soundness:** 3
**Presentation:** 3
**Significance:** 3
**Originality:** 3
**Overall Recommendation:** 5
**Confidence:** 3

**Summary:**

This paper introduces generalized average ranking scores (GARS) based on debiased machine learning using a non-parametric statistical framework DML-Rank for comparing and ranking preferences. GARS generalizes commonly using rankings such as Borda scores, Bradley-Terry model and rank-centrality. The authors demonstrate that using techniques from debiased machine learning results in statistically more efficient estimates (specifically better confidence bounds).

**Compliance With Llm Reviewing Policy:**

Affirmed.

**Final Justification:**

I like the paper and I think it should be accepted in the conference. My evaluation remains unchanged.

**Key Questions For Authors:**

please refer to points in strengths and weakness section.

**Limitations:**

yes

**Strengths And Weaknesses:**

**Strengths**: The authors provide a theoretically sound estimator for GARS using debiased machine learning which naturally yields asymptotic confidence intervals as well. DML provides guarantees even when we use black-box ML to estimate the preference probabilities $\hat \mu$ and the sampling probabilities $\hat \pi$, which are referred to as nuisance parameters. This alleviates the traditional problems of plug-in bias, which propagates to the estimation of ranking scores (GARS). The estimation algorithm is simple and easy to use, and it performs better than the plug-in estimator. Additionally, the authors also empirically demonstrate that the confidence intervals obtained are tight, as shown in Table 1.

Additionally, I also liked the idea of using BT projections as this does not require us to explicitly assume that the rankings come from an underlying BT model.

I checked some of the proofs in the appendices and found them to be correct. However, for a reader not familiar with non-parametric statistics may find this a bit hard to follow. It would be great if the authors could provide right citations at the beginning of Appendix A.

**Weakness**:

1. To my understanding, the authors theoretically study the optimal sampling policy assuming that the preference probabilities are exactly known. While the authors mention in lines 344-364 that it can be computed via an estimate $\hat \mu$, I would like to see how robust it is to mis-estimates either theoretically or empirically. Relatedly, I had the question whether the A-optimal policy computed in Table 2 uses estimates of $\mu$ or its true values.

Is it possible to show how the plug-in bias of $\mu$ distorts the estimate of the optimal sampling policy $\pi^\ast$ (for both A and D optimality) and how does it affect the mse error of the GARS scores estimates?

2. In figure 4(b), BT mis-specification strength is used on the x-axis. However, in its formal definition in appendix J.2.2, the authors use $C$ as the cyclic matrix in its definition. Could you clarify what cyclic matrix means here?

---

> ### Author Rebuttal · Authors · 2026-03-26
>
> Thank you for your positive review and your helpful comments. Below we have drafted careful responses to your suggestions. We will incorporate all points marked with **Action** into the revised version of our paper.
>
> ## Appendix references
>
> Thank you for this suggestion. **Action:** We will add standard references at the beginning of Appendix A on missing-data identification, semiparametric efficiency, and DML / one-step estimation  (e.g., Rubin, van der Vaart, Bickel et al., Chernozhukov et al.), so the setup and proof strategy are easier to follow for readers less familiar with this literature.
>
> ## Estimation errors of $\mu$ in optimal data collection policy
>
> Thank you for bringing up this important point. First, even if $\mu$ is estimated with error, the debiased GARS estimator remains efficient and retains all its favorable properties. We only lose efficiency compared to the (hypothetical) debiased GARS estimator that leverages a different dataset (that is, collected with the optimal policy $\pi$). We used the oracle $\mu$ in Table 2 to illustrate how this can improve over a uniform baseline.
>
> However, we agree that quantifying robustness w.r.t. nuisance estimation is of interest. **Action:** We will add the following result to our paper, which quantifies possible loss in efficiency based on estimation errors in $\mu$.
> Let
> $a_{jk}(x;\mu) =
> tr\Big(J_{jk}(\mu(x))\,V_{jk}(\mu(x))\,J_{jk}(\mu(x))^\top\Big)$,
> and define the A-optimal objective $Q(\pi;\mu) =
> \mathbb{E}\left[\sum_{j\neq k}\frac{a_{jk}(X;\mu)}{\pi_{jk}(X)}\right],
> \qquad \pi\in\Pi_{\alpha,\beta}$.
>
> Let $\pi^\star \in \arg\min_{\pi\in\Pi_{\alpha,\beta}} Q(\pi;\mu)$
> be the oracle A-optimal policy, and let $\hat\pi \in \arg\min_{\pi\in\Pi_{\alpha,\beta}} Q(\pi;\hat\mu)$ be the plug-in A-optimal policy computed from an estimate $\hat\mu$.
> For each fixed policy $\pi$, let $\hat\theta_\pi$ denote the debiased GARS estimator based on data collected under $\pi$, using the *known realized propensities* $\pi$.
>
> **Theorem.**
> Assume the conditions of Theorem 1 hold for every $\pi\in\Pi_{\alpha,\beta}$, and that $\pi_{jk}(x)\ge \alpha>0$ for all feasible $\pi$, $x$, and $j\neq k$. Then $AMSE(\hat\theta_\pi) =
> \frac{1}{n}tr\big(\Sigma(\pi)\big)$
> satisfies
> $0
> \le
> AMSE(\hat\theta_{\hat\pi})-AMSE(\hat\theta_{\pi^\star})
> \le
> \frac{2}{n\alpha}
> \mathbb{E}\left[\sum_{j\neq k}\big|a_{jk}(X;\hat\mu)-a_{jk}(X;\mu)\big|\right]$.
>
> In particular, if for some $L_a<\infty$ it holds $|a_{jk}(x;\hat\mu)-a_{jk}(x;\mu)|
> \le
> L_a |\hat\mu-\mu|_\infty
> \quad\forall x,\; j\neq k$,
>
> then whenever $|\hat\mu-\mu|_\infty \le \varepsilon$,
>
> $AMSE(\hat\theta_{\hat\pi})-AMSE(\hat\theta_{\pi^\star})
> \le
> \frac{2K(K-1)L_a}{\alpha n} \varepsilon$.
>
> Hence, using a plug-in estimate $\hat\mu$ to compute the A-optimal policy induces at most a controlled excess first-order MSE.
>
>
> **Proof sketch.**
>
> From the derivation of Theorem 2, for every fixed $\pi$, $tr\big(\Sigma(\pi)\big)=\mathrm{const}+Q(\pi;\mu)$,
> where the constant does not depend on $\pi$. Therefore it is enough to bound
> $Q(\hat\pi;\mu)-Q(\pi^\star;\mu)$. Add and subtract the plug-in objective:
> $Q(\hat\pi;\mu)-Q(\pi^\star;\mu) = \big(Q(\hat\pi;\mu)-Q(\hat\pi;\hat\mu)\big) +
> \big(Q(\hat\pi;\hat\mu)-Q(\pi^\star;\hat\mu)\big) + \big(Q(\pi^\star;\hat\mu)-Q(\pi^\star;\mu)\big)$.
> The middle term is non-positive since $\hat\pi$ minimizes $Q(\cdot;\hat\mu)$. Hence
> $Q(\hat\pi;\mu)-Q(\pi^\star;\mu)
> \le
> 2\sup_{\pi\in\Pi_{\alpha,\beta}} |Q(\pi;\mu)-Q(\pi;\hat\mu)|$.
> Using $\pi_{jk}(x)\ge \alpha$, it holds that $|Q(\pi;\mu)-Q(\pi;\hat\mu)|
> \le
> \frac{1}{\alpha}
> \mathbb{E}\left[\sum_{j\neq k}|a_{jk}(X;\hat\mu)-a_{jk}(X;\mu)|\right]$.
>
> Combining this yields $tr\big(\Sigma(\hat\pi)\big)-tr\big(\Sigma(\pi^\star)\big)
> \le
> \frac{2}{\alpha}
> \mathbb{E}\left[\sum_{j\neq k}|a_{jk}(X;\hat\mu)-a_{jk}(X;\mu)|\right]$.
>
> Finally, since the debiased estimator under each fixed realized policy $\pi$ has first-order MSE
> $AMSE(\hat\theta_\pi)=\frac{1}{n}tr\big(\Sigma(\pi)\big)$,
> the stated bound follows immediately.
>
> ## Clarification of cycle matrix/ BT misspecification.
>
> Thank you for catching this. What we mean `cyclic matrix'' is a fixed matrix $C$ with $C_{jj}=0$ and $C_{jk}=-C_{kj}$ for all $j\neq$  that encodes a directed preference cycle among items. For example, in the three-item case, one may choose $1$ and $-1$ everywhere, which represents the cycle $1 \le 2$, $2 \le3$, and $3 \le 1$. We then define the misspecified preference model as $\mu_{jk}(x)=\sigma\left(d_{jk}(x)+\gamma C_{jk}\right)$, where $d_{jk}(x)$ is the BT-style score difference and $\gamma \ge 0$ controls the strength of the cyclic, non-transitive component. Hence, $\gamma=0$ corresponds to the correctly specified BT case, while larger $\gamma$ induces stronger deviations from BT transitivity.
> Action: We will add an explicit definition of $C$ in Appendix J.2.2 and a brief example in the main-text discussion of Fig. 4(b), so that “BT misspecification strength” is immediately interpretable.

---

> > ### Author Rebuttal · Reviewer_1SSE · 2026-04-03
> >
> > Thank you for the rebuttal—it addresses my concerns. I have one follow-up question: you introduce a class of cyclic matrices in the rebuttal. In the simulations, are these matrices chosen adversarially, or are they sampled (e.g., uniformly) at random?

---

> > > ### Author Response · Authors · 2026-04-03
> > >
> > > Thank you for your positive feedback! We sincerely appreciate the effort you spent in reviewing our paper.
> > >
> > > Regarding the remaining question: the cycle matrix in our experiments is fixed to a matrix only containing 1 and -1 (thus inducing a preference cycle). What varies is the parameter $\gamma$, which quantifies the strength of the cycle component and thus the degree of BT misspecification (this is the parameter which we plot over).

---

### Decision · Program_Chairs · 2026-04-30

**Decision:**

Accept (regular)

**Comment:**

Motivated by ranking-model misspecification, this paper proposes GARS, a unified semiparametric framework for ranking from pairwise comparison data. GARS defines ranking targets as functionals of nonparametrically estimated pairwise comparison probabilities. This allows the authors to apply debiased machine learning techniques to derive efficient estimators with uncertainty quantification, incorporate black-box comparison oracles, and derive an optimal label-acquisition rule. Reviewers were positive and recommended acceptance, citing the technical soundness, clear methodological motivation, and relevance beyond LLM evaluation. The rebuttal addressed several concerns on exposition, novelty framing, and additional experiments.

At the same time, the AC notes several important issues that should be addressed. While the paper argues that existing estimators can be misspecified or statistically inadequate, it does not sufficiently substantiate this through comparisons against standard practical baselines such as BT MLE, empirical Borda, or classical Rank Centrality, which leaves the practical usefulness of GARS under-motivated. Similarly, the optimal data-acquisition contribution is not compared adequately against relevant line of work on optimal design for preference elicitation, which includes Mukherjee et al. (NeurIPS 2024) and its follow-up work. In addition, the exposition of GARS remains somewhat unclear, especially regarding the motivation and derivation of some proposed functional formulations, and the DML assumptions are not carefully verified for the ranking functionals considered. The authors are strongly encouraged to address these issues and incorporate the discussion with the reviewers in the next revision.

Mukherjee, S., Lalitha, A., Kalantari, K., Deshmukh, A., Liu, G., Ma, Y. and Kveton, B., 2024. Optimal design for human preference elicitation. Advances in Neural Information Processing Systems, 37, pp.90132-90159.